# Statin prevents cancer development in chronic inflammation by blocking interleukin 33 expression

Jong Ho Park[1,2,3], Mahsa Mortaja[1,2], Heehwa G. Son[1,2], Xutu Zhao [1,2], Lauren M. Sloat [1,2], Marjan Azin[1,2], Jun Wang[2], Michael R. Collier[4], Krishna S. Tummala[5,6,7,8,9], Anna Mandinova[2], Nabeel Bardeesy [5,6,7,8], Yevgeniy R. Semenov[4,10], Mari Mino-Kenudson [11] & Shadmehr Demehri [1,2,4] ✉

Chronic inflammation is a major cause of cancer worldwide. Interleukin 33 (IL-33) is a critical initiator of cancer-prone chronic inflammation; however, its induction mechanism by environmental causes of chronic inflammation is unknown. Herein, we demonstrate that Toll-like receptor (TLR)3/4-TBK1-IRF3 pathway activation links environmental insults to IL-33 induction in the skin and pancreas inflammation. An FDA-approved drug library screen identifies pitavastatin to effectively suppress IL-33 expression by blocking TBK1 membrane recruitment/activation through the mevalonate pathway inhibition. Accordingly, pitavastatin prevents chronic pancreatitis and its cancer sequela in an IL-33-dependent manner. The IRF3-IL-33 axis is highly active in chronic pancreatitis and its associated pancreatic cancer in humans. Interestingly, pitavastatin use correlates with a significantly reduced risk of chronic pancreatitis and pancreatic cancer in patients. Our findings demonstrate that blocking the TBK1-IRF3-IL-33 signaling axis suppresses cancer-prone chronic inflammation. Statins present a safe and effective prophylactic strategy to prevent chronic inflammation and its cancer sequela.

Chronic inflammation accounts for 20% of cancers worldwide[1–3]. Cancer-prone chronic inflammation, such as pancreatitis, inflammatory bowel disease (IBD), and hepatitis, have risen in recent decades, highlighting the urgent need for improved cancer prevention strategies in at-risk populations[4–6]. For instance, chronic pancreatitis is associated with an eight-fold increase in the risk of pancreatic cancer 5 years after diagnosis[7]. Several immune cells and factors, including M2

macrophages, mast cells, transforming growth factor (TGF)-β, interleukin 10 (IL-10), and IL-13, have been identified to promote carcinogenesis in chronic inflammation[8–12]. However, inhibiting these effectors alone or combined to prevent cancer has proven challenging due to their redundant function in cancer promotion[13–15]. Furthermore, anti-inflammatory medications, including dexamethasone, which can reduce the risk of cancer development by broadly suppressing

[1]Center for Cancer Immunology, Krantz Family Center for Cancer Research, Massachusetts General Hospital and Harvard Medical School, Boston, MA, USA. [2]Cutaneous Biology Research Center, Department of Dermatology, Massachusetts General Hospital and Harvard Medical School, Boston, MA, USA. [3]Department of Anatomy, School of Medicine, Keimyung University, Daegu, South Korea. [4]Department of Dermatology, Massachusetts General Hospital and Harvard Medical School, Boston, MA, USA. [5]Krantz Family Center for Cancer Research, Massachusetts General Hospital and Harvard Medical School, Boston, MA, USA. [6]Center for Regenerative Medicine, Massachusetts General Hospital, Boston, MA, USA. [7]Department of Medicine, Harvard Medical School, Boston, MA, USA. [8]Cancer Program, Broad Institute of Massachusetts Institute of Technology and Harvard, Cambridge, MA, USA. [9]Quantitative Biosciences, Merck Research Laboratories, Boston, MA, USA. [10]Laboratory of Systems Pharmacology, Harvard Program in Therapeutic Science, Harvard Medical School, Boston, USA. [11]Department of Pathology, Massachusetts General Hospital and Harvard Medical School, Boston, MA, USA. ✉e-mail: sdemehri1@mgh.harvard.edu

immune responses[16], have severe side effects, including coagulopathy and immune hyper-activation, limiting their use as cancer-preventive agents[17,18]. To overcome these challenges, it is essential to develop safe agents that can block the development of chronic inflammation and thereby prevent its cancer sequela.

IL-33 is an epithelium-derived alarmin cytokine that is a member of the IL-1 cytokine family and drives type 2 immune responses in allergic inflammation by triggering T helper 2 (Th2) cell and type 2 innate lymphoid cell (ILC2) activation[19,20]. IL-33 is a critical initiator of chronic inflammation[21,22]. By binding to its receptor, suppressor of tumorigenesis 2 (ST2, also known as interleukin-1 receptor-like 1 (IL1RL1)), IL-33 promotes the development of a chronic inflammatory environment in stressed and inflamed tissues[20]. IL-33 and ST2 are highly expressed in chronic inflammatory diseases, including IBD, pancreatitis, hepatitis, and chronic obstructive pulmonary disease[21-28]. IL-33 plays a complex role in cancer development. IL-33 induction suppresses colon tumor growth and activates CD8+ T and natural killer (NK) cells to inhibit lung metastasis in mice[29,30]. In contrast, the upregulation of IL-33 during the transition from acute to chronic inflammation initiates the development of a tumor-promoting immune environment[21,31-33]. Notably, IL-33 also acts as a nuclear protein and promotes tumorigenesis by regulating SMAD signaling in chronic inflammation[22]. Thus, blocking IL-33 expression instead of its cytokine function alone is essential to achieve cancer prevention in chronic inflammation.

Chronic inflammation is triggered by innate recognition of damage-associated molecular patterns (DAMPs) and pathogen-associated molecular patterns (PAMPs) in injured tissues[34,35]. Mitogen-activated protein kinase (MAPK), Nuclear factor-κB (NF-κB), and TANK-binding kinase 1 (TBK1) signaling are the cardinal intracellular pathways activated by DAMPs and PAMPs upon binding to pattern recognition receptors (PRRs), including Toll-like receptors (TLRs), retinoid acid-inducible gene I (RIG-I)-like receptors (RLRs), and nucleotide-binding oligomerization domain (NOD)-like receptors (NLRs)[34,36-38]. NF-κB, activator protein 1 (AP-1), cAMP-response element binding protein (CREB), interferon regulatory factor 3 (IRF3), and IRF7 are among the major transcription factors activated by PRRs, which can regulate downstream inflammatory cytokines[39-44]. Several of these signaling pathways can induce IL-33 expression[45,46]. RLRs and CREB signaling induce IL-33 expression in mouse embryonic fibroblasts and macrophage cell lines[45]. However, the mechanism of IL-33 induction in epithelial cells during the development of chronic inflammation remains unknown.

Herein, we identify TLR3/4-TBK1-IRF3 signaling as the critical regulator of IL-33 expression and discover statin to suppress IL-33 expression by regulating TBK1 signaling in cancer-prone chronic inflammation. TLR3/4-TBK1-IRF3 signaling was highly activated in the skin and pancreas chronic inflammation, and knockdown of IRF3 blocked *Il33* expression in vitro and in vivo. Pitavastatin, identified from an FDA-approved drug library screening, reduced membrane-bound phosphorylated TBK1 (p-TBK1) through mevalonate pathway inhibition, which suppressed p-IRF3 and IL-33 expression. Accordingly, pitavastatin reduced the risk of pancreatitis and PDAC in mice and humans. We conclude that blocking the TBK1-IRF3-IL-33 axis by statin represents an actionable strategy to prevent chronic inflammation and its cancer sequela.

## Results

### Chronic inflammatory insults in the skin and pancreas activate the TLR3/4 signaling pathway

To determine the mechanism of IL-33 induction in chronic inflammation, we subjected wild-type (WT) mice to established models of chronic inflammation in the skin and pancreas[21,22,47,48]. To induce chronic dermatitis, WT mice received topical 2,4-Dinitro-1-fluorobenzene (DNFB, a contact allergen) in acetone or acetone alone

(carrier control) on the back skin three times a week for 22 days. Epidermal thickness and mast cell numbers were significantly increased in DNFB-treated skin (Supplementary Fig. 1a, b). To induce chronic pancreatitis, mice received intraperitoneal caerulein injections in phosphate-buffered saline (PBS) or PBS alone hourly for 6 hours per day, three days per week for three weeks[47,48]. Caerulein treatment led to inflammation and fibrosis in the pancreas, associated with a significant CD45+ leukocyte infiltration (Supplementary Fig. 1c, d). Consistent with previous reports[23,24], IL-33 was highly expressed in the epithelial cells of DNFB-treated skin and caerulein-treated pancreas compared with acetone-treated skin and PBS-treated pancreas, respectively (Fig. 1a). IL-33 RNA and protein levels were significantly increased in the inflamed compared with control tissues (Supplementary Fig. 2a–d). IL-33 co-localized with cytokeratin in epithelial cells of inflamed skin and pancreas tissues (Supplementary Fig. 2e). To identify the signaling pathway that induced IL-33 in chronic inflammation, we performed RNA sequencing on the epidermal keratinocytes isolated from the back skin of WT mice treated with DNFB versus acetone and the pancreas of WT mice treated with caerulein versus PBS. Among differentially expressed genes, nine common genes were increased in DNFB-treated skin and caerulein-treated pancreas (Supplementary Fig. 2f, g). Among them, *S100a8* and *S100a9*, well-known DAMPs and TLR 3/4 ligands[49,50], were highly enriched in chronic inflammation. Consistently, Gene Set Enrichment Analysis (GSEA) revealed the activation of the TLR3/4 signaling pathway in chronic pancreatitis (Fig. 1b, c). Thus, TLR3/4 signaling may induce IL-33 expression in chronic inflammation.

### TRIF-mediated TBK1-IRF3 signaling regulates IL-33 expression

To determine the nature of TLR3/4 signaling in chronic inflammation, we examined the activation of TLR3/4 downstream targets, TRIF, and MyD88 adaptor proteins[34]. TRIF-mediated TBK1 and IRF3 phosphorylation (i.e., activated forms of TBK1 and IRF3) were markedly increased in DNFB-treated skin and caerulein-treated pancreas compared with acetone and PBS-treated controls, respectively (Fig. 1d, e and Supplementary Fig. 3a, b). Likewise, the known IRF3 target genes, *Tnf*, *Il1b*, and *Cxcl10*, were significantly induced in DNFB-treated skin and caerulein-treated pancreas (Supplementary Fig. 3c–h). However, MyD88-mediated NF-κB (p65) signaling was not activated in either chronic inflammatory condition. To determine whether *Il33* expression was downstream of TBK1-IRF3 signaling, we treated epithelial cells with polyinosinic-polycytidylic acid (poly(I:C)), a TLR3 agonist, and Lipopolysaccharides (LPS), a TLR4 agonist[45,51]. Poly(I:C) and LPS significantly increased *Il33* expression in a mouse keratinocyte cell line, Pam212 (Fig. 1f). Likewise, poly(I:C) increased phosphorylated TBK1 and IRF3 in Pam212 cells after 6 hours and markedly induced the known IRF3 target genes expression (Fig. 1g and Supplementary Fig. 3i–k). Knocking down *Trif* expression by siRNA blocked poly(I:C)-induced *Il33* expression (Fig. 1h). *Il33* promoter region contained a sequence very similar to IRF3 binding motif (mm10;chr19+:29944807) (Fig. 1i, bottom). Importantly, poly(I:C) promoted the binding of phosphorylated IRF3 to the *Il33* promoter region (Fig. 1i, top and Supplementary Fig. 3l), which was abrogated after knocking down *Irf3* by siRNA (Supplementary Fig. 3m). In addition, *Irf3* knockdown markedly reduced poly(I:C)-induced *Il33* expression (Fig. 1j). These findings demonstrate that IL-33 is a downstream target of TRIF-mediated TBK1-IRF3 signaling.

### IRF3 is required for the induction of IL-33 and chronic inflammation in the skin and pancreas

To determine whether *Il33* expression is mediated by TBK1-IRF3 signaling in chronic inflammation, we subjected *Irf3* knockout (Irf3KO) mice to chronic inflammatory conditions in the skin and pancreas. IL-33 RNA and protein levels were significantly reduced in DNFB-treated Irf3KO compared with WT skin (Fig. 2a, b). Moreover, epidermal

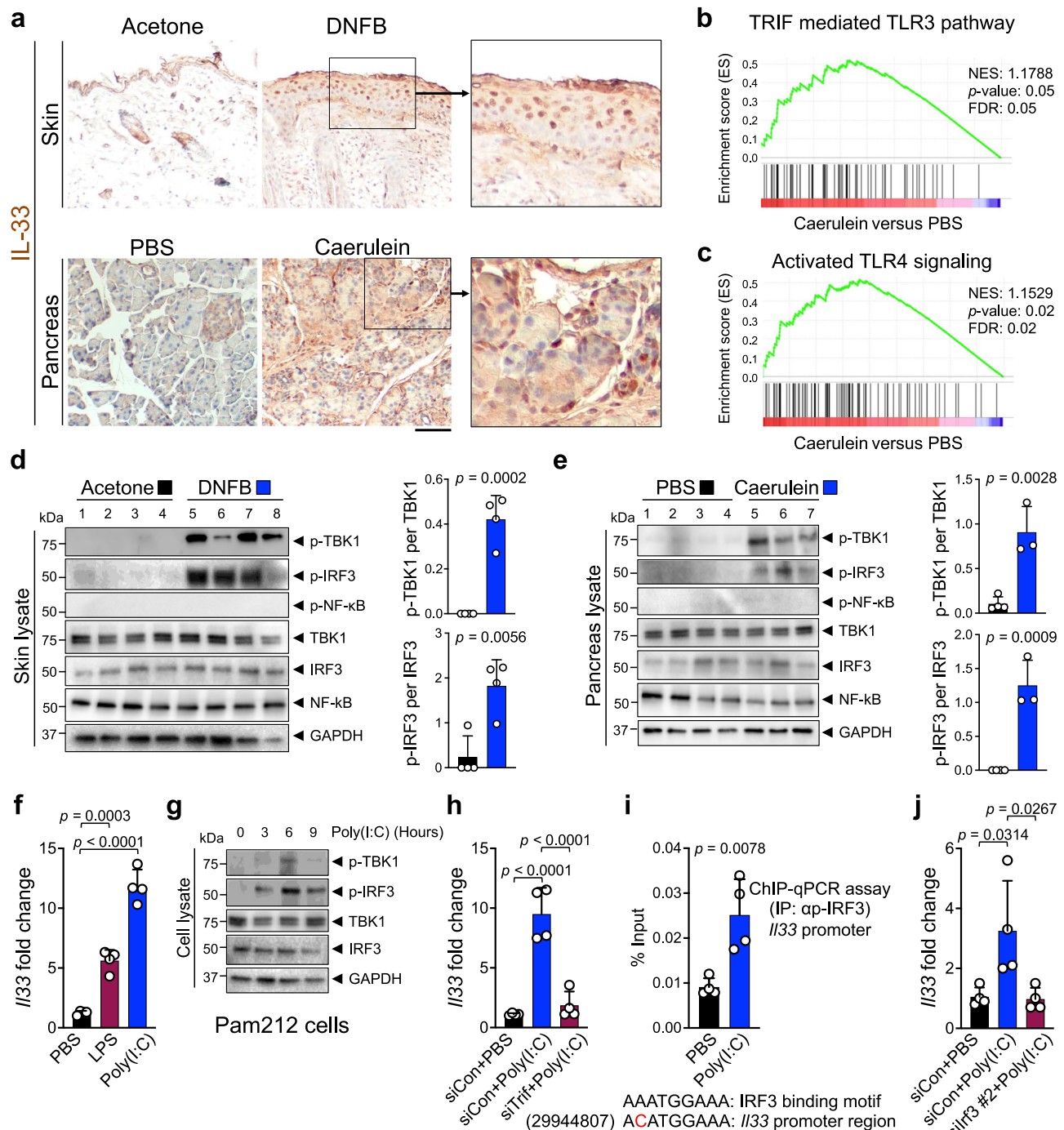

**Fig. 1 | TBK1-IRF3 signaling induces IL-33 expression in chronic inflammation.**
**a** Representative images of IL-33 immunostaining on DNFB-treated skin and
caerulein-treated pancreas compared with the acetone- and PBS-treated controls,
respectively. Note the nuclear IL-33 stains in the epithelial cells of the inflamed
organs. **b** The enrichment plot of TRIF mediated TLR3 signaling gene set from
differentially expressed gene list of caerulein- compared with PBS-treated pancreas
($n = 5$ mice in each group, Kolmogorov-Smirnov test). **c** The enrichment plot of
activated TLR4 signaling gene set from differentially expressed gene list of caer-
ulein- compared with PBS-treated pancreas ($n = 5$ mice in each group, Kolmogorov-
Smirnov test). **d** (Left) Immunoblot of p-TBK1, p-IRF3, p-NF-κB (p65), TBK1, IRF3,
NF-κB and GAPDH proteins in DNFB- versus acetone-treated WT skin ($n = 4$ mice in
each group). (Right) The ratio of p-TBK1/TBK1 and p-IRF3/IRF3 protein band
intensity quantified from the immunoblot shown on the left. **e** (Left) Immunoblot of
p-TBK1, p-IRF3, p-NF-κB, TBK1, IRF3, NF-κB and GAPDH proteins in caerulein- versus
PBS-treated WT pancreas ($n = 3$ mice in caerulein and $n = 4$ mice in PBS group).

(Right) The ratio of p-TBK1/TBK1 and p-IRF3/IRF3 protein band intensity quantified
from the immunoblot shown on the left. **f** *Il33* expression in Pam212 cells at 6 hours
after treatment with poly(I:C), LPS and PBS ($n = 4$ cell culture plates in each group).
**g** Time course of p-TBK1, p-IRF3, TBK1, IRF3 and GAPDH protein expression in
Pam212 cells after poly(I:C) treatment. **h** *Il33* expression in *Trif* siRNA (siTrif)- versus
control siRNA (siCon)-treated Pam212 cells in response to poly(I:C) versus PBS
treatment ($n = 4$ cell culture plates in each group). **i** Chromatin Immunoprecipita-
tion (ChIP)-qPCR assay for p-IRF3 binding to *Il33* promoter region after the treat-
ment of Pam212 cells with poly(I:C) versus PBS ($n = 4$ cell culture plates in each
group). Note the presence of a sequence very similar to the IRF3 binding motif in
the *Il33* promoter region. **j** *Il33* expression in siIrf3- versus siCon-treated Pam212
cells in response to poly(I:C) versus PBS treatment ($n = 4$ cell culture plates in each
group). Graphs show mean + SD, d, e, i: two-sided unpaired *t*-test, f, h, j: one-way
ANOVA, scale bar: 100 μm. Source data are provided as a Source Data file.

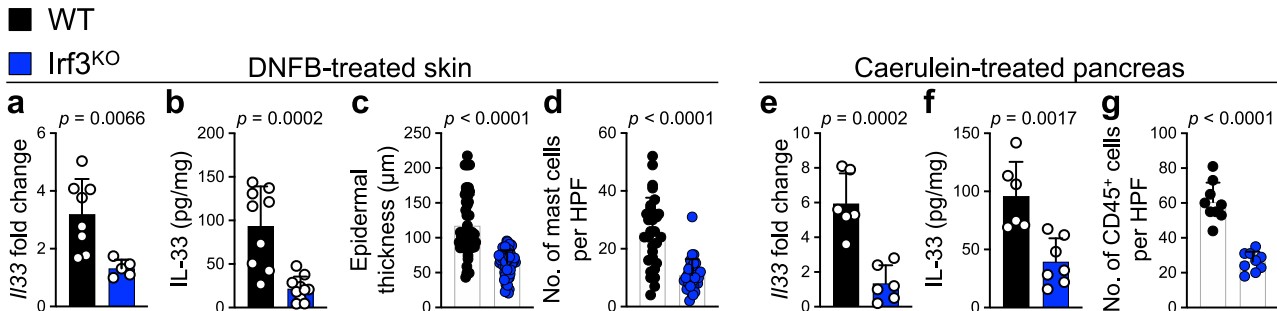

**Fig. 2 | IRF3 activates IL-33 expression in chronic inflammation. a** *Il33* expression levels in DNFB-treated WT versus Irf3[KO] skin ($n = 8$ in WT and $n = 5$ mice in Irf3[KO] group). **b** IL-33 protein levels in DNFB-treated WT versus Irf3[KO] skin ($n = 9$ mice in WT and $n = 10$ mice in Irf3[KO] group). **c** Epidermal thickness in DNFB-treated WT versus Irf3[KO] skin. Each dot represents the average of three measurements in a high-power field (HPF) image. Ten random HPF images per skin sample are included ($n = 6$ mice in each group). **d** Mast cell counts in DNFB-treated WT versus Irf3[KO] skin. Each dot represents cell counts from an HPF image. Three randomly selected HPF images are included per skin sample ($n = 10$ mice in each group). **e** *Il33* expression levels in caerulein-treated WT versus Irf3[KO] pancreas ($n = 6$ mice in each group). **f** IL-33 protein levels in caerulein-treated WT versus Irf3[KO] pancreas ($n = 6$ mice in WT and $n = 7$ mice in Irf3[KO] group). **g** CD45[+] immune cell counts in caerulein-treated WT versus Irf3[KO] pancreas. Each dot represents cell counts from an HPF image. Three randomly selected HPF images are included per mouse pancreas ($n = 3$ mice in each group). Graphs show mean + SD, two-sided unpaired *t*-test. Source data are provided as a Source Data file.

thickness and mast cell numbers were decreased markedly in DNFB-treated Irf3[KO] compared with WT skin (Fig. 2c, d and Supplementary Fig. 4a, b). Consistent with these results, IL-33 levels were reduced in DNFB-treated skin of *Trif* and *Myd88* double knockout (Trif,Myd88[DKO]) mice but not in Myd88[KO] mice (Supplementary Fig. 4c), which indicates that the TRIF adapter protein is the primary upstream activator of the IRF3 signaling pathway to induce IL-33 in chronic inflammation. IL-33 RNA and protein levels decreased in caerulein-treated Irf3[KO] compared with WT pancreas (Fig. 2e, f). Likewise, caerulein-treated Irf3[KO] pancreas showed less inflammation and reduced CD45[+] leukocyte infiltration compared with WT pancreas (Fig. 2g and Supplementary Fig. 4d). Collectively, these findings demonstrate that TBK1-IRF3 regulates IL-33 expression in chronic dermatitis and pancreatitis.

### Pitavastatin blocks TBK1 phosphorylation and IL-33 expression via mevalonate pathway inhibition

To identify a small molecule IL-33 inhibitor that can safely alleviate chronic inflammation and its cancer sequela, we screened an FDA-approved drug library in a luciferase-based *Il33* expression assay (Supplementary Fig. 5a, b). Among 1018 FDA-approved small molecules that were screened, we found five candidates which decreased *Il33*/control luminescence absorbance to less than 40% while not affecting absorbance in a control luminescence assay (Supplementary Fig. 5c). Among these candidates, pitavastatin calcium (labeled as O16 in the screen) suppressed poly(I:C)-induced *Il33* expression in Pam212 cells (Supplementary Fig. 5d). Pitavastatin also suppressed endogenous *Il33* expression in PyMt[tg] breast cancer cell line, which had high *Il33* expression at baseline (Supplementary Fig. 5e). Pitavastatin is a lipophilic statin that inhibits β-Hydroxy β-methylglutaryl-CoA (HMG-CoA) reductase, an intermediate reaction in the mevalonate pathway[52]. Pitavastatin and zoledronic acid, a bisphosphonate that inhibits mevalonate pathway but does not affect HMG-CoA reductase, equally suppressed poly(I:C)-induced *Il33* expression, while a TBK1 inhibitor, BX795, completely blocked poly(I:C)-induced *Il33* expression in Pam212 cells (Fig. 3a). Interestingly, lipophilic statins, pitavastatin and atorvastatin, inhibited *Il33 expression* more potently compared with a hydrophilic statin, rosuvastatin (Supplementary Fig. 5f). Reduced efficacy of rosuvastatin in blocking *Il33* expression may reflect its reduced uptake into the cells due to its hydrophilic nature.

Statins inhibit HMG-CoA reductase, which reduces geranylgeranyl diphosphate (GGPP), a product of the mevalonate pathway[52,53]. GGPP plays a critical role in the membrane localization of intracellular proteins[54]. Mevalonate pathway inhibition by pitavastatin blocked poly(I:C)-induced activation of the TBK1-IRF3 signaling pathway in

Pam212 cells (Fig. 3b). Importantly, poly(I:C) treatment led to the recruitment of TBK1 to the membrane for phosphorylation (i.e., activation), and pitavastatin markedly reduced membrane-bound p-TBK1 (Fig. 3c). The addition of exogenous GGPP to Pam212 cells reversed pitavastatin effect and restored TBK1-IRF3 signaling pathway activation and membrane-bound p-TBK1 levels (Fig. 3b, c). To validate these findings, we examined the effect of pitavastatin on the TBK1-IRF3 pathway in a pancreatic cell line. Pitavastatin suppressed TBK1-IRF3 signaling pathway activation and membrane-bound p-TBK1 levels in the pancreatic cells, and this effect was also reversed by exogenous GGPP (Supplementary Fig. 6a, b). Consistent with these findings, poly(I:C)-induced p-TBK1 co-localized with an endoplasmic reticulum (ER) membrane marker, calreticulin (Supplementary Fig. 6c)[55,56]. Similar to pitavastatin, zoledronic acid reduced membrane-bound p-TBK1 in poly(I:C)-treated Pam212 cells, which was reversed by exogenous GGPP (Supplementary Fig. 6d). Considering that pitavastatin and zoledronic acid target proteins are different, these results indicate that mevalonate pathway inhibitors suppress TBK1 signaling. Accordingly, pitavastatin suppression of poly(I:C)-induced *Il33* expression was reversed by exogenous GGPP (Fig. 3d). HMG-CoA reductase RNA (*Hmgcr*) levels were not affected by pitavastatin and GGPP treatment (Supplementary Fig. 7a). Moreover, adding exogenous cholesterol to Pam212 cells did not reverse the effect of pitavastatin (Supplementary Fig. 7b). Thus, pitavastatin inhibits *Il33* expression by blocking GGPP-dependent membrane recruitment and activation of TBK1 independent of cholesterol levels (Fig. 3e). GGPP-mediated prenylation plays a crucial role in membrane targeting of proteins and signal transduction[57–59]. We examined whether TBK1 regulation by GGPP is related to prenylation. Interestingly, GGTI-2147, an inhibitor of prenylation[60,61], blocked TBK1 membrane localization and phosphorylation (Supplementary Fig 7c). Therefore, TBK1 may be a target of GGPP-mediated prenylation.

### Pitavastatin suppresses chronic inflammation and its cancer sequela in an IL-33-dependent manner

Next, we investigated the impact of pitavastatin treatment on suppressing IL-33 and chronic inflammation in vivo. To test the pitavastatin effect on skin inflammation, mice were treated with topical DNFB on the back skin for 22 days together with topical pitavastatin versus carrier control (acetone). Pitavastatin treatment reduced p-TBK1 and p-IRF3 levels in the skin compared with acetone-treated mice (Supplementary Fig. 8a). Accordingly, IL-33 RNA and protein levels were markedly decreased in pitavastatin- compared with acetone-treated mice (Supplementary Fig. 8b, c). Skin inflammation, as marked by

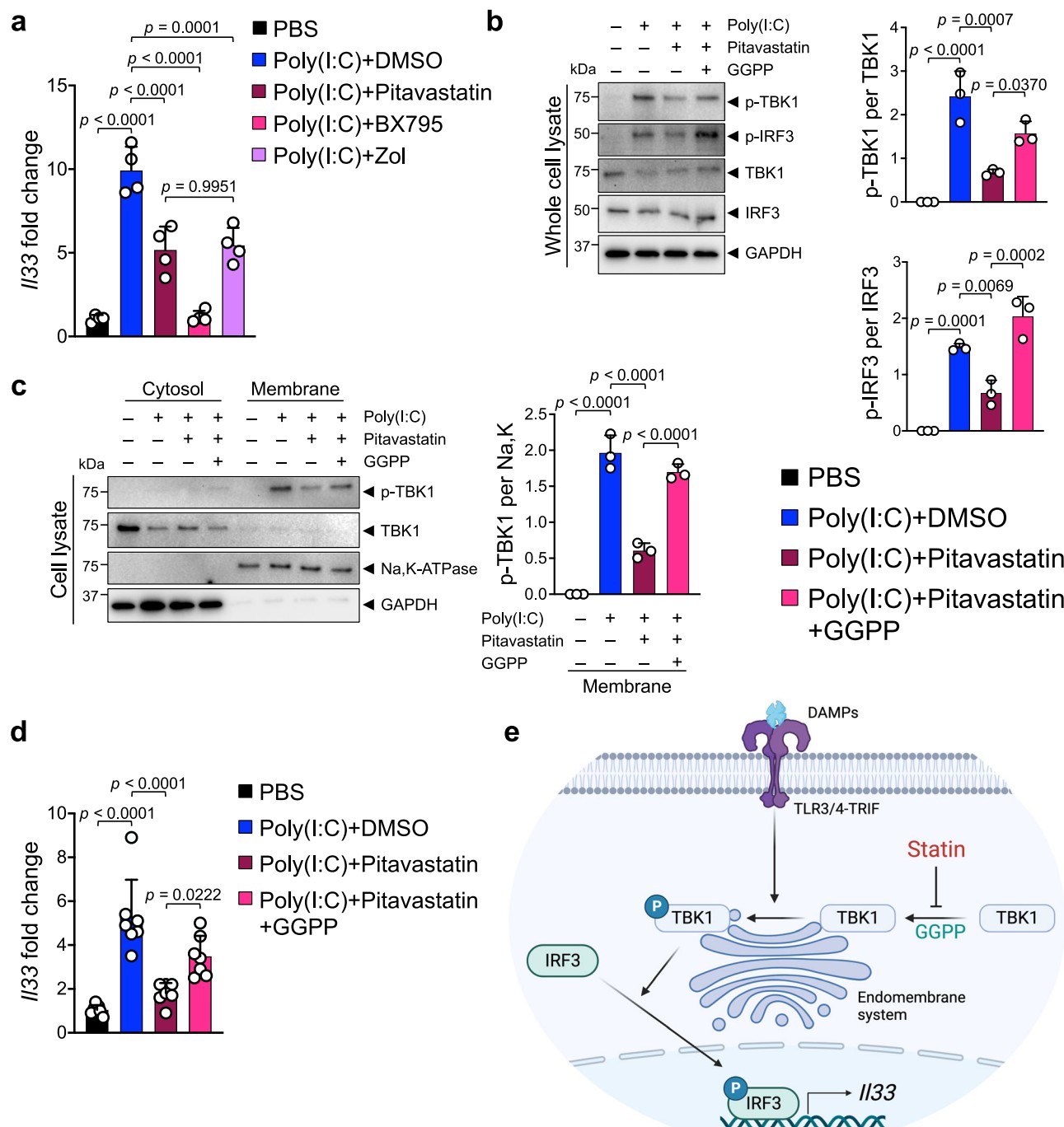

**Fig. 3 | Pitavastatin inhibits IL-33 expression by blocking GGPP-dependent TBK1 activation. a** *Il33* expression in poly(I:C)-treated Pam212 cells that received pitavastatin (10 µM), BX795 (10 µM), Zoledronic acid (Zol) (10 µM) or PBS (*n* = 4 cell culture plates in each group). Cells were harvested after 6-hour incubation with poly(I:C) and each inhibitor. **b** (Left) Immunoblot of p-TBK1, p-IRF3, TBK1, IRF3, and GAPDH proteins in whole cell lysates of poly(I:C)-treated Pam212 cells that received pitavastatin alone (10 µM) or in combination with GGPP (3 µM). (Right) The ratio of p-TBK1/TBK1 and p-IRF3/IRF3 protein band intensity from immunoblots (*n* = 3 cell culture plates in each group). Cells were harvested after 6-hour incubation with poly(I:C), pitavastatin, and GGPP. **c** (Left) Immunoblot of p-TBK1, TBK1, Na-K-ATPase, and GAPDH proteins in membrane and cytosol fraction of poly(I:C)-treated Pam212 cells that received pitavastatin (10 µM) alone or in combination with GGPP (3 µM). Cells were harvested after 6-hour incubation with poly(I:C), pitavastatin, and GGPP. (Right) The ratio of membrane-bound p-TBK1/Na,K-ATPase protein band intensity from the immunoblots (*n* = 3 cell culture plates in each group). **d** *Il33* expression in poly(I:C)-treated Pam212 cells that received pitavastatin (10 µM) alone or in combination with GGPP (3 µM) (*n* = 7 cell culture plates in each group). Cells were harvested after 6-hour incubation with poly(I:C), pitavastatin, and GGPP. **e** Schematic diagram of pitavastatin mechanism of action in blocking mevalonate pathway-GGPP mediated TBK1-IRF3 signaling activation (created with BioRender.com). Graphs show mean + SD, one-way ANOVA. Source data are provided as a Source Data file.

epidermal thickness and mast cell numbers in the skin, was significantly reduced in pitavastatin- compared with acetone-treated mice (Supplementary Fig. 8d-f). To examine the effect of pitavastatin on chronic pancreatitis, mice were treated with caerulein for three weeks together with intraperitoneal pitavastatin versus carrier control (PBS). Pitavastatin treatment significantly reduced p-TBK1 and p-IRF3 levels in the pancreas with no effect on NF-κB signaling (Fig. 4a). Likewise, pitavastatin significantly reduced IL-33 RNA and protein

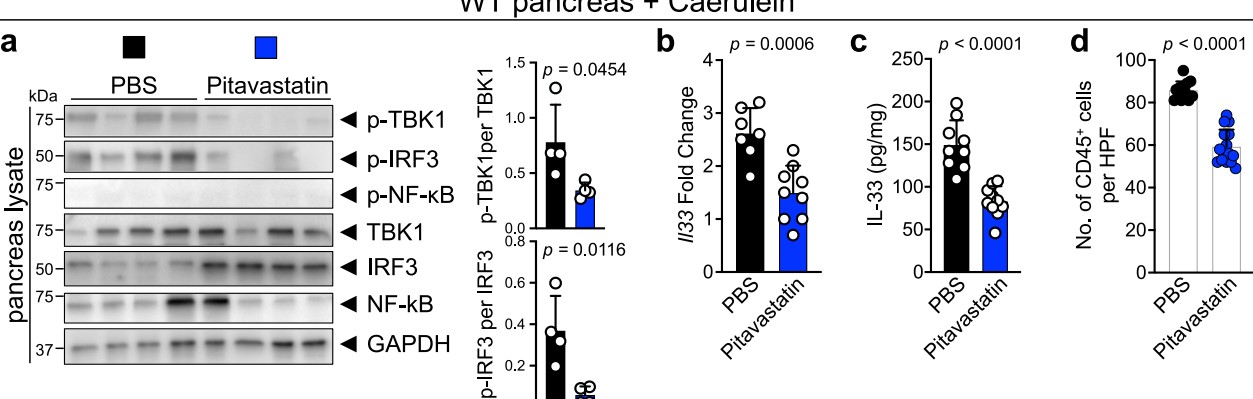

**Fig. 4 | Pitavastatin suppresses chronic pancreatitis by blocking the TBK1-IRF3-IL-33 signaling axis. a** (Left) Immunoblot of p-TBK1, p-IRF3, p-NF-κB, TBK1, IRF3, NF-κB and GAPDH proteins in pitavastatin- versus PBS-treated WT pancreas at the completion of caerulein treatment protocol (n = 4 mice in each group). (Right) The ratio of p-TBK1/TBK1 and p-IRF3/IRF3 protein band intensity quantified from the immunoblot shown on the left. **b** *Il33* expression in pitavastatin- versus PBS-treated WT pancreas at the completion of caerulein treatment protocol (n = 9 mice in pitavastatin and n = 7 mice in PBS group). **c** IL-33 protein levels in pitavastatin-

versus PBS-treated WT pancreas at the completion of caerulein treatment protocol (n = 10 mice in pitavastatin and n = 9 mice in PBS group). **d** CD45+ immune cell counts in pitavastatin- versus PBS-treated WT pancreas at the completion of the caerulein treatment protocol. Each dot represents cell counts from an HPF image. Three randomly selected HPF images are included per mouse pancreas (n = 5 mice in pitavastatin and n = 4 mice in the PBS group). Graphs show mean + SD, two-sided unpaired t-test. Source data are provided as a Source Data file.

levels in the caerulein-treated pancreas (Fig. 4b, c). Moreover, pitavastatin treatment preserved the normal architecture of the caerulein-treated pancreas and reduced CD45+ leukocyte infiltration into the pancreas compared with PBS-treated mice (Fig. 4d and Supplementary Fig. 9a). Importantly, pitavastatin had no significant impact on the severity of pancreatitis in Il33[KO] mice (Supplementary Fig. 9b, c). Thus, pitavastatin prevents chronic inflammation by suppressing the TBK1-IRF3-IL-33 signaling axis in vivo.

Chronic pancreatitis is a risk factor for the development of pancreatic cancer[62,63]. To establish a chronic pancreatitis-associated pancreatic cancer model, we treated pancreas-specific *Kras* and *Tp53* mutant (*Kras*[LSL-G12D], *Tp53*[flox/+], *p48-Cre*[tg] or KPC) mice with hourly intraperitoneal injections of caerulein for 7 hours per day for two consecutive days (Supplementary Fig. 9d)[64–66]. This pancreatic carcinogenesis protocol led to a significant induction of IL-33 expression in the pancreas, including pancreatic epithelial cells (Supplementary Fig. 9e, f). KPC mice were treated with pitavastatin versus PBS control after the 2nd caerulein injection. Pitavastatin 4-week treatment significantly reduced pancreatic tumor weight per body weight ratio compared with PBS-treated KPC mice (Fig. 5a, b). Moreover, pitavastatin treatment blocked the progression of pancreatic tumors and retained the tumor cells in a pre-cancerous stage with high mucin production compared with PBS-treated tumors (Fig. 5a, c). Notably, pitavastatin long-term treatment increased the survival rate of KPC mice compared with PBS-treated controls and retained the pancreatic tumors in a pre-cancerous stage (Fig. 5d and Supplementary Fig. 9g, h). In contrast, there was no significant difference in pancreatic tumor per body weight ratio or mucin production by tumor cells in Il33[KO] KPC mice treated with pitavastatin versus PBS control (Fig. 5e–g). These findings support the model that pitavastatin blocks chronic pancreatitis-associated pancreatic cancer in an IL-33-dependent manner.

### IRF3-IL-33 axis is highly active in chronic pancreatitis and pancreatic cancer in humans

To extend our findings to cancer-prone chronic inflammation in humans, we examined IL-33 and IRF3 expression in the epithelial cells across 15 matched samples of the normal pancreas, pancreatitis, and pancreatitis-associated pancreatic ductal adenocarcinoma (PDAC). IL-33 and IRF3 were highly expressed in the nucleus of epithelial cells in

pancreatitis and pancreatitis-associated PDAC samples (Fig. 6a–c). Moreover, the number of IL-33+ epithelial cells was positively correlated with the number of IRF3+ epithelial cells across the samples (Fig. 6d). Expression of *IL33* and other IRF3 target genes, *TNF*, *IL1B*, and *CXCL10*, were highly upregulated in pancreatic cancer compared to the normal pancreas in a large collection of samples represented in TCGA and GTEx databases (Fig. 6e and Supplementary Fig. 10).

### Pitavastatin treatment is associated with reduced risk of chronic pancreatitis and pancreatic cancer in patients

Finally, we investigated the effect of pitavastatin on the risk of chronic pancreatitis and pancreatic cancer in humans using an epidemiologic approach. We compared matched cohorts of patients from the Tri-NetX Diamond Network, a global health network containing electronic medical record-derived data from more than 200 million patients across 92 healthcare organizations in North America and Europe (Supplementary Table 1)[67]. The risk of chronic pancreatitis was significantly decreased in patients treated with pitavastatin compared to those treated with ezetimibe, another cholesterol-lowering agent commonly used in the clinic, which does not affect the mevalonate pathway (control, OR 0.81; 95% CI (0.729–0.9); p < 0.0001). Furthermore, the risk of pancreatic cancer was markedly decreased in the pitavastatin-treated group compared with the ezetimibe-treated control (OR 0.835; 95% CI (0.748–0.932); p = 0.0013) (Fig. 6f). Collectively, these outcomes indicate that blocking TBK1-IRF3-IL-33 signaling axis by statins may prevent chronic inflammation and its cancer sequela in high-risk patients.

### Discussion

Our findings reveal that lipophilic statins suppress cancer-prone chronic inflammation by blocking the TBK1-IRF3-IL-33 signaling axis induced by chronic exposure to environmental insults. Cellular damage and release of DAMPs lead to TLR3/4-mediated activation of the TBK1-IRF3 signaling pathway. Phosphorylated IRF3 directly binds to the *Il33* promoter to drive IL-33 expression during the initiation of chronic inflammation. Importantly, pitavastatin blocks *Il33* expression by inhibiting the mevalonate pathway-mediated TBK1 binding to the membrane, which may be required for its phosphorylation and downstream IRF3 activation. By inhibiting *Il33* expression, pitavastatin blocks the cytokine and nuclear functions of IL-33 in chronic

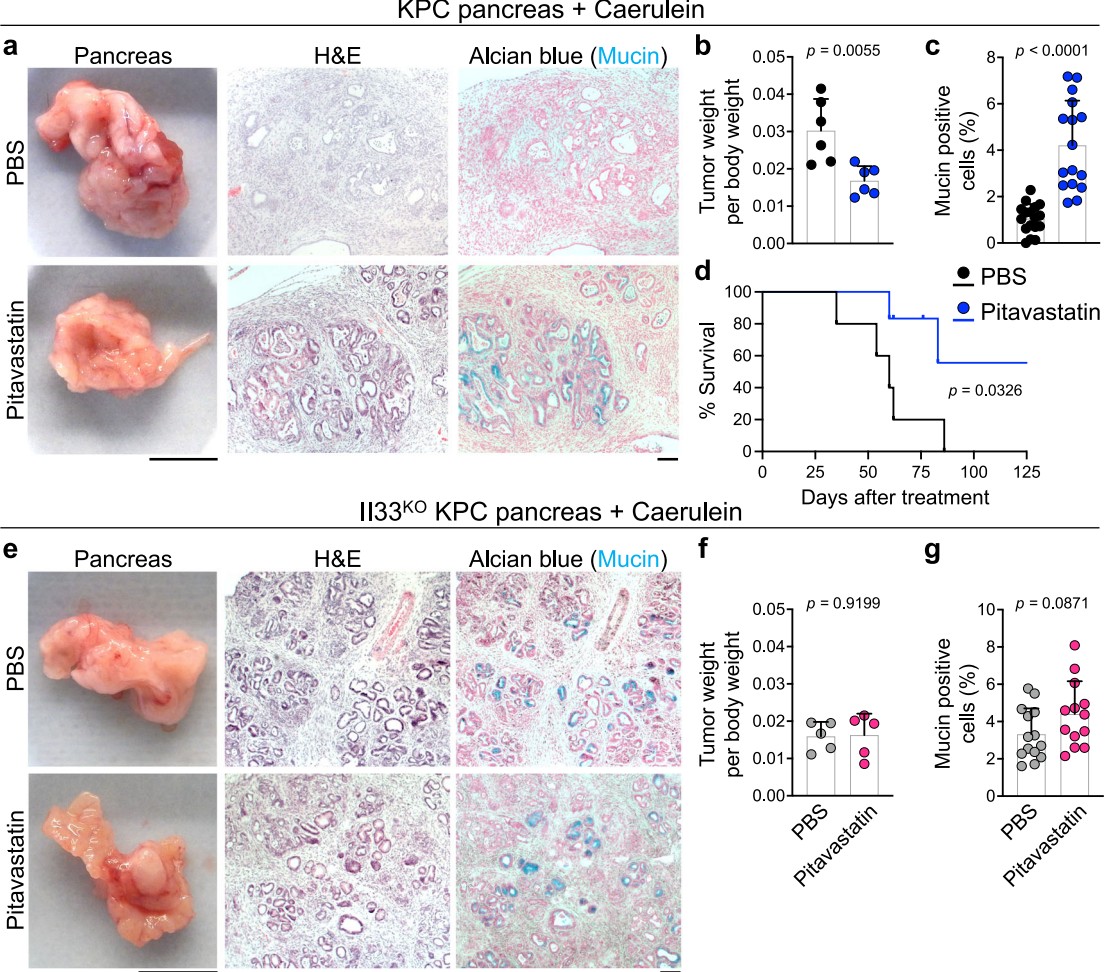

**Fig. 5 | Pitavastatin blocks chronic pancreatitis-associated pancreatic cancer.**
**a** Representative images of macroscopic, hematoxylin and eosin (H&E), and Alcian blue-stained pancreatic tumors from pitavastatin- versus PBS-treated KPC mice that underwent caerulein-induced pancreatic cancer protocol. **b** Ratio of pancreatic tumor per body weight in pitavastatin- versus PBS-treated KPC mice at the completion of 4-week caerulein-induced pancreatic cancer protocol ($n = 6$ mice in each group). **c** Percent mucin-positive cells in pitavastatin- versus PBS-treated KPC tumors at the completion of 4-week caerulein-induced pancreatic cancer protocol. Each dot represents % mucin-positive cells in an HPF image. Two to three randomly selected HPF images are included per mouse pancreas ($n = 6$ mice in each group). **d** Survival of caerulein-exposed KPC mice treated long-term with pitavastatin ($n = 6$ mice) versus PBS ($n = 5$ mice, log-rank test). **e** Representative images of macroscopic, H&E, and Alcian blue-stained pancreatic tumors from pitavastatin- versus PBS-treated Il33$^{KO}$ KPC mice that underwent caerulein-induced pancreatic cancer protocol. **f** Ratio of pancreatic tumor per body weight in pitavastatin- versus PBS-treated Il33$^{KO}$ KPC mice at the completion of 4-week caerulein-induced pancreatic cancer protocol ($n = 5$ mice in each group). **g** Percent mucin-positive cells in pitavastatin- versus PBS-treated Il33$^{KO}$ KPC tumors at the completion of 4-week caerulein-induced pancreatic cancer protocol. Each dot represents % mucin-positive cells in an HPF image. Two to three randomly selected HPF images are included per mouse pancreas ($n = 5$ mice in each group). Graphs show mean + SD, two-sided unpaired $t$-test, scale bars: 1 cm or 100 µm. Source data are provided as a Source Data file.

inflammation, effectively reducing the risk of chronic pancreatitis and pancreatic cancer in mice and humans. Therefore, blocking the TBK1-IRF3-IL-33 signaling axis with statins represents a safe, effective, and readily accessible strategy to prevent chronic inflammation and its cancer sequela, which can impact many individuals at high risk of developing cancer-prone chronic inflammation.

IL-33 has a diverse role across multiple cancer types. The anti-tumor function of IL-33 has been reported in myeloma and colorectal cancer models[68,69]. However, IL-33 pro-tumor function has been noted in multiple cancers, including glioma, gastric, and colorectal cancers[70–72]. In response to intratumoral mycobiome, IL-33 activates type 2 immune response in the tumor microenvironment and promotes pancreatic cancer development[73]. Furthermore, IL-33 plays a pro-tumorigenic role as a nuclear protein by regulating SMAD signaling and GATA3, independent of its cytokine function[22,74]. Thus, the diverse effects of IL-33 on cancer may depend on its localization in the tissue, including epithelial cells, fibroblasts, versus immune cells[73,75–77].

Because IL-33 is expressed in epithelial cells of the skin and pancreas during chronic inflammation, statin's efficacy in cancer suppression may be related to its ability to block nuclear and cytokine functions of IL-33 in cancer-initiating epithelial cells. Statin may also block IL-33 pro-tumor function in immunocompromised settings, which needs to be examined in future research.

Several malignancies are associated with activated TBK1-IRF3 signaling pathway within the cancer cells, which can play a critical cell-autonomous role in cancer progression[78,79]. In particular, TBK1 activation has been linked to skin and pancreatic cancer development[80,81]. Furthermore, high TBK1 expression has been shown to induce an immunosuppressive tumor microenvironment by increasing PD-L1 expression and inhibiting CD8$^+$ T cell infiltration in lung, liver cancer, and melanoma[82–84]. Accordingly, TBK1 inhibitors have tumor inhibitory effects associated with improved sensitivity to immunotherapy in several cancer types, including melanoma and liver cancer[81,82,84,85]. Moreover, activated TBK1-IRF3 signaling leads to the induction of

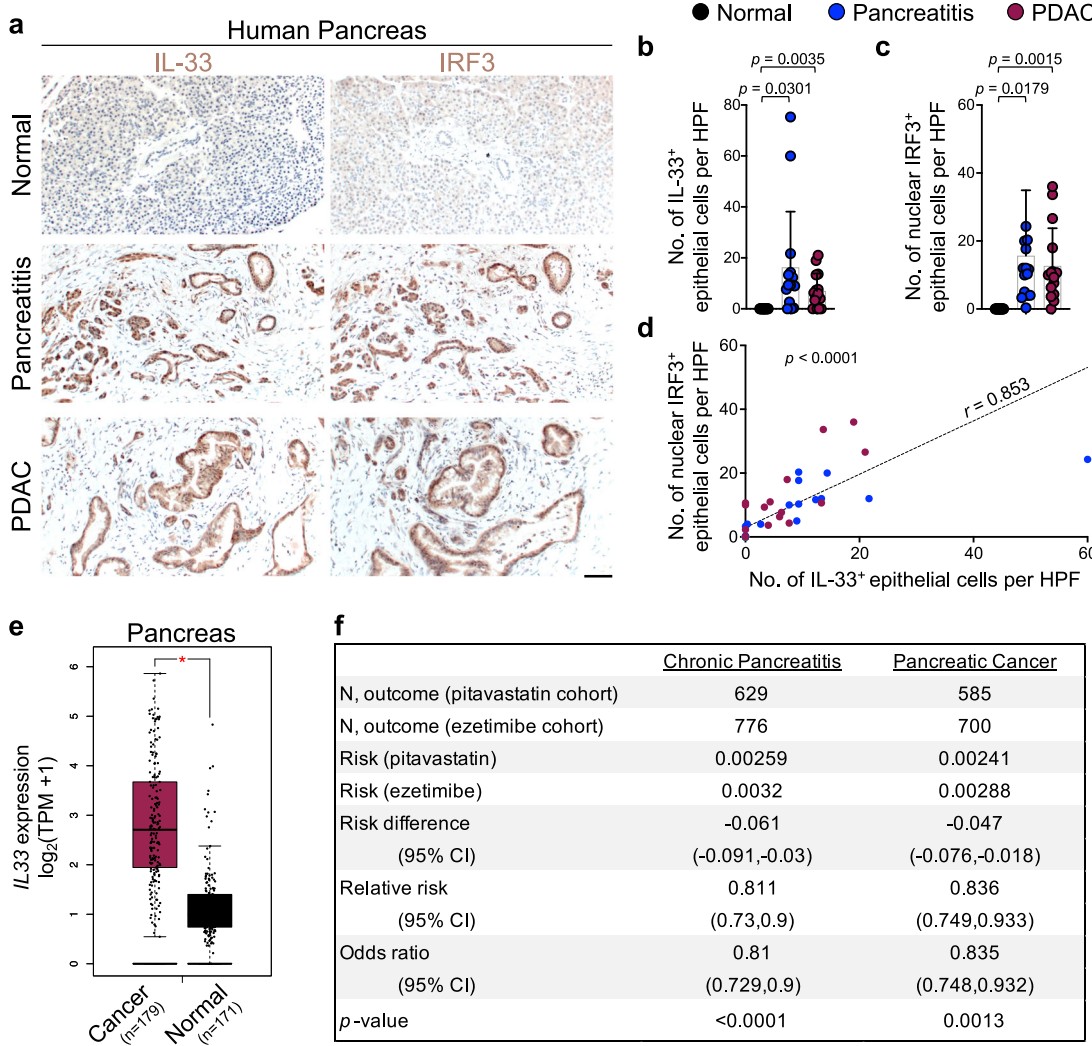

**Fig. 6 | IRF3-IL-33 signaling axis is highly active in human chronic pancreatitis-associated pancreatic cancer, and pitavastatin reduces the risk of chronic pancreatitis and pancreatic cancer in patients. a** Representative images of IL-33 and IRF3 immunostaining on adjacent sections of matched normal pancreas, chronic pancreatitis, and PDAC collected from pancreatic cancer patients. **b** IL-33⁺ epithelial cell counts per HPF in the matched samples from human pancreatic tissues. Each dot represents the average cell counts across three randomly selected HPF images per sample (*n* = 15 patients, one-way ANOVA). **c** IRF3⁺ epithelial cell counts per HPF in the matched samples from human pancreatic tissues. Each dot represents the average cell counts across three randomly selected HPF images per sample (*n* = 15 patients, one-way ANOVA). **d** The correlation between IL-33⁺ and IRF3⁺ cell counts across normal pancreas, chronic pancreatitis, and PDAC samples (*n* = 45 samples from 15 patients, two-sided *t*-test for the Pearson correlation coefficient). **e** Box plot of *IL33* expression in pancreatic cancer versus normal pancreas across TCGA/GTEx datasets (*$p < 0.01$, one-way ANOVA, Gene Expression Profiling Interactive Analysis database). **f** A retrospective cohort analysis of chronic pancreatitis and pancreatic cancer risk in matched cohorts of patients treated with pitavastatin (test) versus ezetimibe (control) (two-sided two-proportion z-test). Bar graphs show mean + SD, box plots show median (center line), interquartile range (box limits), 1.5 times the interquartile range (whiskers length), and outliers (points outside whiskers), scale bar: 100 μm. Source data are provided as a Source Data file.

angiogenesis factors associated with poor prognosis in several cancers, including pancreatic cancer[79,86–88]. Previous studies have identified IRF3 as a regulator of IL-33 in vitro[51,89]; however, we demonstrate that IL-33 is the target of TBK1-IRF3 signaling activation in cancer-prone chronic inflammation. Likewise, IL-33's pro-tumor function can play an integral role in tumor promotion by TBK1-IRF3 signaling[31–33]. Thus, targeting the TBK1-IRF3-IL-33 signaling axis is an attractive strategy to block cancer development.

The mevalonate pathway has recently emerged as an important regulator of cancer development. Blocking the mevalonate pathway product, GGPP, has been shown to inhibit cancer cells' amino acid uptake by regulating macropinocytosis[90]. Moreover, the mevalonate pathway inhibition by statins induces a robust antitumor T cell immunity by promoting antigen presentation[54]. Importantly, our findings reveal a previously unknown mechanism by which the mevalonate pathway regulates TBK1-IRF3 signaling through its essential role

in the membrane recruitment of TBK1, which may be required for its phosphorylation/activation. Our findings also provide a potential explanation for how GGPP regulates PI3K/MAPK pathway activation by promoting the membrane localization of the critical signaling molecules in the pathway[91,92]. Thus, elucidating statins' effect on IL-33 inhibition by targeting TBK1-IRF3 signaling pathway may have significant implications in revealing a fundamental aspect of TBK1-IRF3 and other intracellular signaling pathways.

Our work uncovers an unprecedented role for statins as a safe class of chemopreventive agents for suppressing chronic inflammation and its cancer sequela. Statins are commonly used for long-term control of hyperlipidemia, and 56 million adults are taking statins in the United States alone[93]. Statins are well-tolerated over many years of treatment with minimal side effects[94], which signifies their high potential as effective agents for cancer prevention. Unlike TBK1 inhibitors that are in development with a high cost and potential side

effects, statins are affordable FDA-approved medications that can be safely prescribed for long-term use, which are essential requirements for an ideal chemopreventive agent. Furthermore, our findings highlight the topical application of statins as an innovative treatment strategy for cancer-prone chronic inflammation in the skin. Likewise, statins may present a breakthrough for pancreatic cancer prevention. Pancreatic cancer is an insidious cancer type known for its unresponsiveness to current treatments, including immunotherapies, due to its highly immunosuppressive tumor microenvironment with dense desmoplastic stroma[95,96]. Although statins' impact on an established tumor microenvironment is context-dependent[97–101], our findings strongly indicate that statin use can block the development of cancer-prone chronic inflammation and the formation of an immunosuppressive tumor microenvironment in high-risk patients. Importantly, statin use is associated with an increased survival rate in pancreatic cancer patients[102,103]. Moreover, in a large population study, we demonstrate that pitavastatin markedly reduces the risk of pancreatic cancer development compared with ezetimibe. Finally, the beneficial effect of statins in blocking the TBK1-IRF3-IL-33 axis may also extend to other IL-33-dependent chronic inflammatory conditions, including chronic obstructive pulmonary disease (COPD), atopic dermatitis, and asthma[104–106].

## Methods

### Study approval
Massachusetts General Hospital IRB approved the analysis of de-identified clinical tissue samples. Massachusetts General Hospital IACUC approved the animal studies. The condition of mice in experiments was permitted and monitored by our facility members.

### Human samples
De-identified formalin-fixed paraffin-embedded human pancreas tissue sections were obtained from the Department of Pathology at Massachusetts General Hospital. IRB approved the study protocol (using de-identified tissue samples) and waived informed consent in our case because we only obtained de-identified tissue sections from the Department of Pathology for our studies. Patients did not receive any compensation as part of this study.

### Animal studies
All mice were housed under pathogen-free conditions on regular chow in an animal facility at Massachusetts General Hospital in accordance with animal care regulations. Irf3[KO] mice were purchased from the Riken BioResource Research Center (Ibaraki, Japan). Il33[KO] mice were a gift from Dr. Marco Colonna, and Tp53[flox] (*Trp53^{tm1Brnn}*/J), p48-Cre[tg] (*Ptf11a^{tm1(cre)Hnak}*/RschJ), Kras[LSL-G12D] (*Kras^{tm4Tyj}*/J), Myd88[KO] (*Myd88^{tm1.1Defr}*/J), Trif[KO] (*Ticam1^{Lps2}*/J) and C57BL/6 WT mice were purchased from the Jackson Laboratory (Bar Harbor, ME). With regard to cancer studies, Massachusetts General Hospital IACUC permitted a 20 mm maximal tumor diameter, and the maximal tumor size was not exceeded. Mice were euthanized under anesthesia at predetermined time points or following humane endpoints, including tumors reaching 20 mm in diameter, weight loss greater than 20% of the initial total body weight, a weight increase of 20% due to ascitic fluid, a hunchbacked appearance, or a moribund state.

### Skin chronic inflammation
Four- to six-week-old male and female mice were shaved on their abdomen and sensitized to 50 µL 0.5% 1-Fluoro-2,4-dinitrobenzene (DNFB, Millipore Sigma, St. Louis, MO, catalog no. 42085) dissolved in acetone with olive oil at 3:1 ratio (refer to as acetone). Two days after the first sensitization, mice were sensitized to 50 µL 0.25% DNFB on their abdomen again. After five days, mice were challenged with 100 µL 0.25% DNFB on their back skin, which was repeated thrice weekly for 22 days. Skin rash was monitored over the duration of the study.

### Chronic pancreatitis
Mice were weighed and injected with 50 µg/kg caerulein (BACHEM, Torrance, CA, catalog no. 4030451) in 100 µL of PBS intraperitoneally every hour for 6 hours, three days per week for three weeks. Mice were harvested for analysis after the three-week treatment protocol.

### Caerulein-mediated pancreatic cancer
Mice were weighed and injected with 50 µg/kg caerulein in 100 µL of PBS intraperitoneally every hour for 7 hours for two consecutive days. Mice were harvested 30 days after the last injection or followed long-term for survival analysis.

### Pitavastatin treatment
For chronic inflammation in the skin, mice were treated topically with 0.25 mM pitavastatin (Selleck Chemicals LLC, Houston, TX, catalog no. S1759) in 200 µL acetone or 200 µL acetone alone on their back skin twice a week. Pitavastatin treatments were given at the time of DNFB applications. For chronic pancreatitis and caerulein-mediated pancreatic cancer, a high concentration of pitavastatin in DMSO (5 mg/mL) was prepared as a stock solution. The stock solution was diluted with PBS to avoid DMSO toxicity in mice. The mice were treated intraperitoneally with 2 mg/kg pitavastatin delivered in 100 µL PBS or the same volume of DMSO plus PBS (referred to as PBS) for the control group based on previous studies[107–109]. Pitavastatin was given once every three days until harvest for pancreatitis and 4-week pancreatic cancer studies. For the pancreatic cancer survival study, KPC mice received pitavastatin or PBS once a week after the 4-week once every three days treatment. The intraperitoneal route for pitavastatin treatment was chosen as it is an established and safer route compared with oral gavage for repeated administration of drugs in mice[107–109].

### Cell lines and transfection
Pam212 keratinocyte cell lines and 839WT pancreatic cells were grown at 37 °C in DMEM (Thermo Fisher Scientific, Waltham, MA, catalog no.11995065) supplemented with 10% fetal bovine serum (Thermo Fisher Scientific, catalog no. 26140079), 1X penicillin-streptomycin-glutamine (Thermo Fisher Scientific, catalog no. 10378016), 1X MEM non-essential amino acids solution (Thermo Fisher Scientific, catalog no. 11140050), 1X HEPES (Thermo Fisher Scientific, catalog no. 15630080), and 0.1% 2-Mercaptoethanol (Thermo Fisher Scientific, catalog no. 21985023). 839WT pancreatic cells were generated by adapting 3D organoids derived from the pancreas of WT mice on the C57BL/6 background to a 2D culture system. This was achieved by culturing the organoids with mitomycin-treated NIH3T3 fibroblasts and gradually acclimating them to grow without additional growth factors on a Matrigel-coated petri dish. PyMt[tg] cell line (derived from a primary breast tumor of MMTV-PyMT[tg] mouse on a C57BL/6 background) was grown at 37 °C in RPMI Medium 1640 (Thermo Fisher Scientific, catalog no. 11875093) supplemented with 10% fetal bovine serum, 1X penicillin-streptomycin-glutamine, and 0.1% 2-Mercaptoethanol. Transfections of LPS (Millipore Sigma, catalog no. L6529) or poly(I:C) (Invivogen, San Diego, CA, catalog no. tlrl-pic) in Pam212 and 839WT cells were performed using Lipofectamine 2000 (Thermo Fisher Scientific, catalog no. 11668019) according to the manufacturer's instructions. For gene knockdown, Pam212 cells were transfected with siRNA construct using Lipofectamine RNAiMax Transfection reagent (Thermo Fisher Scientific, catalog no. 13778075) according to the manufacturer's instructions. Specific siRNA constructs are listed in Supplementary Table 2.

### Stable cell lines
After transfection of mouse *Il33* promoter clone (GeneCopoeia, Rockville, MD, catalog no. MPRM34949-PG02) or mouse control promoter clone (deleted promoter region from mouse *Il33* promoter clone) with Lipofectamine 2000 in Pam212 cells, cells were incubated

with 3 μg/mL of puromycin (Invivogen, catalog no. ant-pr-1) for positive transfected cell selection. After two days, the medium was changed with a new puromycin-containing medium for further selection. After the second two days, cells were transferred to a new plate with a puromycin-containing medium. Puromycin selection was repeated five more times. Then, cells were passaged in regular media to ensure optimal growth.

### Small compound screening with Luciferase assay

One thousand stable cells were seeded in 384 wells (Corning, Glendale, AZ, catalog no. 3570), and each plate was incubated for 24 hours. The following day, 10 μM of FDA-approved Drug Library compounds (Selleckchem, catalog no. L1300, 2019 version) were added to each well. After 24 hours, 5 μL of luciferase buffer from Pierce Gaussia luciferase glow assay kit (Thermo Fisher Scientific, catalog no.16160) was added to each well and incubated for 5 min. Each plate was measured with luminescence by an EnVision 2014 plate reader (Perkin Elmer, Waltham, MA, catalog no. 2014 EnVision). The screening assay was performed in duplicate. A TBK1 inhibitor, BX795, was used as a positive control in IL-33 suppression assays.

### Western blot

Cell lysates were prepared in LIPA buffer (Thermo Fisher Scientific, catalog no. 89900) consisting of 1X protease inhibitor cocktail, EDTA-free (Thermo Fisher Scientific, catalog no. A32955). Mice tissues were meshed and lysed by 0.1% TWEEN-20 (Millipore Sigma, catalog no. P1379) in PBS. Mice tissues were frozen in liquid nitrogen and thawed by incubation at 37 °C for further lysis. Tissue lysates were sonicated for 10-20 seconds and centrifugated at 23,000 g. After checking the protein concentration in each sample, the identical amounts of total proteins were loaded onto Mini-PROTEIN TGX™ Gels (BIO-RAD, Hercules, CA, catalog no.456-1083 or 456-1086) with 1X Tris/Glycine/SDS buffer (BIO-RAD, catalog no.1610732). A few minutes later (according to protein size, ~30 min with 200 Voltage), they were transferred to Immobilon−P membrane (Millipore Sigma, catalog no. IPVH00010) with Transfer buffer (Boston Bioproducts, Ashland, MA, catalog no. BP-190). Then, the samples were incubated with 3% bovine serum albumin (Thermo Fisher Scientific, catalog no. BP1600) or 5% Skim milk (BD biosciences, San Jose, CA, catalog no. 232100) in 1X Tris-Buffered Saline (Boston Bioproducts, catalog no. BM301X) containing 0.1% TWEEN, called TBS-T for 30 min. After washing with TBS-T three times, the membranes were subjected to immunoblot with proper antibodies at 4 °C overnight. The following day, the membranes were incubated with appropriate secondary antibodies after washing. Membranes were developed with Pierce ECL Western blotting substrate kit (Thermo Fisher Scientific, catalog no. 32106). First and secondary antibodies are listed in Supplementary Table 2.

### Western blot quantification

Western blot bands were quantified with cSeries Capture software (Azure, Biosystem, Bulin, CA, catalog no. Azure 600) and BIO-RAD (BIO-RAD, catalog no. 1700140). Each band quantity was calculated by measuring band intensity minus background. Total proteins were normalized based on GAPDH levels. p-TBK1 and p-IRF3 were measured as the ratio of total endogenous TBK1 and IRF3 levels, respectively, after confirming that endogenous TBK1/GAPDH and IRF3/GAPDH levels were not altered between the comparison groups.

### Histology, immunohistochemistry, and immunofluorescence

Tissue samples were collected and fixed in 4% paraformaldehyde (Millipore Sigma, catalog no. P6148) at 4 °C overnight. Next, tissues were dehydrated in PBS and ethanol, processed, and embedded in paraffin. Five to seven μm sections of paraffin-embedded tissues were placed on slides, deparaffinized, and stained with H&E, toluidine blue (for mast cell) (Millipore Sigma, catalog no. T3260), or Alcian blue (for

mucin) (VECTOR Laboratories, Burlingame, CA, catalog no. H3501). For immunohistochemistry, antigen retrieval was performed in 500 μL of antigen unmasking solution (VECTOR Laboratories, catalog no. H3300) diluted in 50 mL distilled water using a Cuisinart pressure cooker for 20 min at high pressure. Slides were washed three times for three minutes each in 1X TBS with 0.025% Triton X-100. Sections were blocked with 3% bovine serum albumin (Thermo Fisher Scientific, catalog no. BP1600) and 5% goat serum (Millipore Sigma, catalog no. G9023) for 1 hour. Slides were incubated at 4 °C overnight with a primary antibody diluted in TBS containing 3% BSA (Supplementary Table 2). The following day, slides were washed as above and incubated in 100 μL biotinylated secondary antibody (VECTOR Laboratories, catalog no. PK-6200) for 30 min. After washing, slides were stained with a 100 μL mixture of reagents A and B from VECTASTAIN Elite ABC universal kit Peroxidase (VECTOR Laboratories, catalog no. PK-6200) for 30 min. After washing again, slides were incubated with 100 μL ImmPACT DAB chromogen staining (VECTOR Laboratories, catalog no. SK-4105) for a few minutes (depending on the signal). Finally, slides were dehydrated in ethanol and xylene and mounted with a coverslip using three drops of mounting media. For immunofluorescence staining, rehydrated tissue sections were permeated with 1X PBS supplemented with 0.2% Triton X-100 for 5 min. Antigen retrieval was performed similarly to immunohistochemistry. Slides were washed three times for 3 min each in 1X PBS with 0.1% Tween-20. Sections were blocked with 3% bovine serum albumin and 5% goat serum for 1 hour. The slides were incubated at 4 °C overnight with primary antibodies. The following day, slides were washed as above and stained for 2 hours at room temperature with secondary antibodies conjugated to fluorochromes. Next, slides were incubated with 1:2,000 DAPI (Thermo Fisher Scientific, catalog no. D3571) for 3 min at room temperature, then washed as above. For cell line immunostaining, Pam212 cells were seeded in the 2-well chambered cell culture slides (CELLTREAT, Pepperell, MA, catalog no. 229162) in 2 mL media. After incubation with poly(I:C) versus PBS for 6 hours, cells were fixed with 4% paraformaldehyde and stained with anti-pTBK1 antibody using Alexa Fluor 488 Tyramide SuperBoost Kit (Thermo Fisher Scientific, catalog no. B40943) and then anti-calreticulin antibody after permeabilization and blocking (Supplementary Table 2). Slides were mounted with Prolong Gold Antifade Reagent (Thermo Fisher Scientific, catalog no. P36930). We used the Tyramide SuperBoost Kit for the same species antibody staining according to the manufacturer's instructions (Thermo Fisher Scientific, catalog no. B40942 or B40943). The number of positive cells was counted in randomly selected high-power field (HPF, 200x magnification) images in a blinded manner by a trained investigator. A pathologist reviewed clinical samples.

### Quantitative PCR

Mouse dorsal skin and pancreas samples were homogenized and lysed in RLT lysis solution (QIAGEN, Hilden, Germany, catalog no. 79216)/ 0.1% MeOH using Mini-BeadBeater-8 (BioSpec Products, Inc., Bartlesville, OK). Trizol reagent was added to tissue samples and cell pellets to extract RNA (Thermo Fisher Scientific, catalog no. 15-596-018). Total RNA was extracted using an RNeasy micro kit and quantified using NanoDrop ND-1100 (NanoDrop Technologies, Wilmington, DE). cDNA was synthesized from 1 μg of total RNA using Invitrogen SuperScripts III Reverse Transcriptase (Thermo Fisher Scientific, catalog no. 18080085). The gene expression levels from cDNA samples were determined using QuantStudio 3 system (Thermo Fisher Scientific) using SYBR Select Master Mix (Thermo Fisher Scientific, catalog no. 4472908) or TaqMan Universal Master Mix II (Thermo Fisher Scientific, catalog no. 44-400-40). Primer sequences for SYBR green and Taqman assays are listed in Supplementary Table 2. Quantitative real-time PCR for SYBR green analyses was performed in a final reaction volume of 20 μL consisting of 5 μL of cDNA of the respective sample and 10 μL of SYBR green master mix mixed with the corresponding primers (2 μM)

for each gene. TaqMan analysis was performed in a 10 μL final reaction, including 4.5 μL cDNA and 5.5 μL TaqMan master mix and corresponding primers (20 μM). All assays were conducted in triplicate and normalized to *Gapdh* expression.

## RNA sequencing

WT mice underwent DNFB-induced skin inflammation and caerulein-induced pancreatitis protocols. For skin RNA sequencing (RNA-Seq), epidermis RNA was isolated from the back skin of five mice in each group. For pancreas RNA-Seq, pancreas RNA was isolated from five mice in each group. Epidermis and pancreas tissues were lysed in TCL buffer (Qiagen, catalog no. 1031576) supplemented with 1% β-mercaptoethanol (Thermo Fisher Scientific, 21-985-023). Libraries were generated and sequenced using the Smart-Seq2 protocol as previously described using Novaseq 6000 (Illumina) on the Broad Genomics Platform[110]. The raw files were mapped to the mouse genome/mm10 by STAR-2.5.3[111]. Aligned transcripts were quantified using RSEM-1.3.1[112]. Differentially expressed genes (DEG) were analyzed by DESEq2[113]. Original data are available in the NCBI Gene Expression Omnibus (GEO) with accession number GSE207956 (RNA-Seq). GSEA analysis was performed with differentially expressed genes between caerulein- and PBS-treated pancreas samples.

## ChIP-qPCR assay

Pam212 cells that had been transfected with poly(I:C) were fixed in 1% formaldehyde (Millipore Sigma, catalog no. F8775) for 10 min and were washed with cold PBS. Cells were lysed with buffer contained with 2.5% of glycerol, 50 mM HEPES (pH 7.5), 150 mM NaCl, 0.5 mM EDTA, 0.5% NP-40, and 0.25% Triton X-100. After centrifugation at 3,000 g, lysates were resuspended in buffer A contained with 1 mM Tris/HCl (pH 7.9), 20 mM NaCl, and 0.5 mM EDTA and incubated at room temperature for 10 min. Following the second centrifugation at 3,000 g, cells were sonicated in a sonication buffer containing 10 mM HEPES, 1 mM EDTA, and 0.5% SDS for 30 minutes to achieve chromatin fragmentation. Following centrifugation at 23,000 g, proteins were immunoprecipitated with the p-IRF3 antibody (Cell Signaling Technology, Danvers, MA, catalog no. 29047) at 4 °C overnight. The following day, the samples were incubated with a 30 μL 50% slurry of ChIP-Grade Protein-G agarose beads (Cell Signaling Technology, catalog no. 9007 S) for 3 hours. Samples were precipitated and washed three times with wash buffer containing 20 mM Tris/HCl (pH 7.9), 10 mM NaCl, 2 mM EDTA, 0.1% SDS, and Triton X-100. After the last wash step, samples were eluted with 100 μL of TE buffer contained with 100 mM Tris/HCl (pH 8), 1 mM EDTA, and 1% SDS three times. De-crosslinking was conducted by overnight incubation with 15 μL 3 M NaCl at 65 °C. Precipitated DNA was eluted by ChIP DNA Clean & Concentrator (ZYMO research, Irvine, CA, catalog no. D5205) and subjected to quantitative PCR. *Il33* promoter sequence was obtained from the GeneCopoeia, Inc. website.

## PCR

DNA was extracted from mouse tissue using KAPA Express Extract buffer and KAPA Express Extract enzyme from Kapa Genotyping kit (Kapa Biosystems Inc, Wilmington, MA, catalog no. KK7302). After tissue lysis, PCR was performed using 2X KAPA2G Fast genotyping mix from the Genotyping kit and the primers listed in Supplementary Table 2.

## Membrane extraction

Pam212 and 839WT cells were transfected with poly(I:C) using Lipofectamine 2000 according to the manufacturer's instructions, and cells were treated with pitavastatin (10 μM), zoledronic acid (10 μM) or GGTI-2147 (5 μM) with or without GGPP (3 μM) (Echelon Biosciences, Salt Lake City, UT, catalog no. I-0200) or cholesterol (5 μg/mL) (Millipore Sigma, catalog no. C8667). After a 6-hour incubation with

poly(I:C) and each small molecule, cells were washed with Cell Wash Solution (Thermo Fisher Scientific, catalog no. 89842) and collected with a cell scraper. After centrifugation at 3,000 g, cell pellets were resuspended in 100 μL of Permeabilized Buffer (Thermo Fisher Scientific, catalog no. 89842) consisting of 1X EDTA-free protease inhibitor cocktail. Samples were incubated for 10 min at 4 °C with constant mixing. After incubation, permeabilized cells were centrifugated for 15 min at 23,000 g, and only the supernatant was carefully transferred to a new tube (Cytosol fraction). Next, the pellets were resuspended in a Solubilization Buffer (Thermo Fisher Scientific, catalog no. 89842) and incubated for 30 min at 4 °C with constant mixing. After incubation, samples were centrifugated for 15 min at 23,000 g, and only the supernatant was transferred to a new tube (membrane fraction).

## TCGA/GTEx RNA expression analysis

RNA-sequencing expression data from pancreatic cancer and normal samples were obtained from the public database: Gene Expression Profiling Interactive Analysis database (GEPIA2) at http://gepia2.cancer-pku.cn/#index[114]. Expression data of 9736 tumors and 8587 normal samples from the TCGA and the GTEx projects were included.

## Epidemiological analysis

A retrospective cohort analysis generated in this study used de-identified data from the TriNetX Diamond Network. A search query was used to identify the cohort of patients within the network who had received pitavastatin. Eligible patients were identified based on the presence of corresponding RxNorm concept unique identifiers (RXCUI) in the patients' electronic medical records. Using International Classification of Diseases Tenth Revision (ICD-10) codes, all patients with a history of chronic pancreatitis and pancreatic cancer prior to statin initiation were excluded from the cohorts to reduce confounding. The control cohort for each analysis included all patients within the network who had received ezetimibe but had no recorded statin use, and patients with a history of any of the diagnoses mentioned above before ezetimibe initiation were also excluded.

The index event for all analyses was the initiation of pitavastatin for study cohorts and the initiation of ezetimibe for the control cohort. Cases and controls were matched using 1:1 propensity score-matching for age at index event, sex, race, and ethnicity using "greedy nearest neighbor matching" and a caliper of 0.1 pooled standard deviations. Baseline characteristics were reported by count and percentage of the total for categorical variables and mean and standard deviations (SD) for continuous variables. Relative risks are presented with 95% confidence intervals. *p*-values are uncorrected and based on Z-tests or Fisher's exact tests. Statistical analyses were performed in real-time using the TriNetX platform.

## Statistical analysis and reproducibility

A paired *t*-test was used for comparing IL-33[+] and IRF3[+] cell counts, and a paired *t*-test for the Pearson correlation coefficient was used for correlation between IL-33[+] and IRF3[+] counts across matched human pancreatic samples. Statistical differences between the three groups were analyzed using one-way ANOVA. Tukey multiple comparison tests were used to examine the differences in the mean ranks among all three possible pairwise comparisons. The risk ratios of epidemiological data were compared using a two-tailed, two-proportion z-test. A two-sided unpaired *t*-test was used to test the significance of tumor per body weight ratio, epidermal thickness, mast cell and leukocyte counts, RNA and protein expression levels, and other quantitative measurements. A *p*-value < 0.05 is considered significant. Bar graphs show mean + SD. Sample sizes were not predetermined based on statistical methods but were chosen according to the standards of the field (at least three independent biological replicates for each condition).

**Reporting summary**

Further information on research design is available in the Nature Portfolio Reporting Summary linked to this article.

## Data availability

All data needed to evaluate the conclusions in the paper are present in the paper and the Supplementary Information. RNA sequencing data can be accessed from the NCBI Gene Expression Omnibus (GEO), accession no: GSE207956. Source data are provided with this paper as a source data file. Source data are provided with this paper.

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

## Acknowledgements

We thank Dr. Marco Colonna for Il33^KO mice. We thank Minyoung Kim for assistance with clinical tissue collection. Il33^KO mice were generated with support from the Mucosal Immunology Studies Team (MIST) (U01; RFA-AI-15-023). S.D. holds a Career Award for Medical Scientists from the Burroughs Wellcome Fund and LEO Foundation Award. J.H.P., M.M., H.G.S., X.Z., L.M.S., M.A., and S.D. were supported by grants from the Burroughs Wellcome Fund, Sidney Kimmel Foundation, and NIH (K08AR068619 and R01AR076013).

## Author contributions

J.H.P. and S.D. conceived and designed the experiments. J.H.P., M.M., H.G.S., X.Z., L.M.S., and M.A. performed the experiments and analyzed the data. J.W. and A.M. contributed to FDA-approved drug library screening. K.S.T. and N.B. contributed pancreatic cell line. M.R.C. and Y.R.S. conducted the epidemiological study. M.M-K. contributed clinical samples. J.H.P. and S.D. interpreted the data and wrote the manuscript.

## Competing interests

J.H.P. and S.D. are coinventors on a filed patent for the use of IL-33 inhibition in treating cancer, fibrosis, and inflammation (PCT/US21/40725). A.M. is an equity holder of DermBiont Inc. The remaining authors state no conflict of interest.
