## [Peer Review File · Nature Communications]

Statin prevents cancer development in chronic inflammation by blocking interleukin 33 expressionREVIEWER COMMENTS

Reviewer #1 (Remarks to the Author):

Park et. al suggest a novel TLR3-TBK1-IRF3 axis induces IL-33 expression during chronic inflammation in skin and pancreas. The authors perform a FDA-approved drug screen and identify pitavastatin as an inhibitor of IL-33 expression by blocking TBK1 membrane localization and activity. They use in vitro and in vivo assays to show the efficacy of pitavastatin inhibition of skin inflammation and chronic pancreatitis as well as in reducing pancreatic tumor progression. In general, the results are intriguing, however several details are missing, mechanism remains unclear, and references to previous studies requires clarification to address novelty. Major and minor issues are listed below.

Major issues:

1. The mechanism of pitavastatin blocking TBK1 membrane localization remains unclear. As they show this inhibition can be reversed with the addition of exogenous GGPP, the evidence shows GGPP is involved in TBK1 membrane localization and activation, but the mechanism remains unclear. GGPP is important for several processes, including i) protein isoprenylation, ii) production of coenzyme Q (ubiquinone) required for electron transport chain activity, and iii) production of dolichol required for protein N-glycosylation. How does GGPP enable TBK1 membrane localization? GGPP-dependent isoprenylation occurs on CAAX-containing proteins. Is TBK1 such a substrate? If so this would suggest a direct effect, which could be shown by an independent approach using GGPP transferase inhibitors to mimic pitavastatin action. In addition, the inhibitory effect of pitavastatin on TBK1 membrane localization could be functionally uncoupled by targeting TBK1 membrane localization by a mevalonate pathway- and GGPP-independent method, e.g. myristoylation of TBK1. If not, then what is the mechanism? This is a major gap that needs to be addressed.
2. The authors invoke that pitavastatin and zoledronic acid are functioning similarly. Yet the mechanism of these drugs is very different, with pitavastatin targeting HMGCR and zoledronic acid targeting bisphosphonates. This needs to be mentioned in the text. Moreover, to support their claim of similar mechanism of action, they need to show that zoledronic acid also leads to mis-localization of TBK1 from membranes and that exogenous GGPP reverses the activity of zoledronic acid. If zoledronic acid and pitavastatin do not behave similarly in action and response to GGPP, then their distinct mechanisms of action need to be described and put into context for the reader to be able to follow. Are these two different inhibitors of the mevalonate pathway functioning by a similar or different mechanism.
3. Membrane localization of TBK1 needs to be validated by an independent assay as the high salt concentrations used during membrane extraction often disrupt localization. The usual method of choice is to monitor TBK1 localization using microscopy and image-based analyses, including immunofluorescence and/or by fluorophore tagging of membrane-bound TBK1.
4. The results are presently impossible to reproduce as many of the key experimental details are missing. Throughout the manuscript the concentration, delivery method and duration of drug treatment need to be provided for each experiment. For example, Figure 2 and Extended data figure 4: Please mention the concentration of pitavastatin, atorvastatin and GGPP used, as well time of treatment. The lack of activity of rosuvastatin may be an uptake issue as it is hydrophobic. This needs to be mentioned in the text as a caveat to data interpretation.
5. The authors have delivered pitavastatin as a topical and intraperitoneal treatment, yet as cholesterol-control agents their FDA-approved delivery is to take them orally. Therefore the authors need to explain why they didn't treat the mice by oral gavage, to mimic the patient experience. They focused on pitavastatin because it is an approved drug, yet they don't use it by the approved route of delivery. This needs to be discussed and justified.
6. Line 225: 'demonstrate' needs to be changed to 'support the model that' as the evidence is only correlative in nature. A direct functional relationship has not been supported by the data provided.
7. Lines 282 and 283: "Blocking the mevalonate pathway product, GGPP, inhibits cancer cells' amino acid uptake by regulating micropinocytosis (ref 66)." This is incomplete as GGPP is important for many downstream processes (see #1 above), which need to be described in the text and addressed mechanistically.

8. Lines 289-291: Overstated. The data provided do not explain mechanism of action.
9. This axis of IL-33 regulation and the inhibitory role of statins has been previously reported, yet references were not included, e.g. PMID: 36711701; PMID: 31810599. Please include these and similar previous references and clearly delineate to the reader what was previously described and what's new in this report.
10. Figure 1D: Please provide the full membrane for the western blot. The image quality looks dissimilar in different parts of the same membrane. Furthermore, the authors show the ratio of p-TBK1/TBK1 and p-IRF3/IRF3 quantified from immunoblot. However, they should quantify also related to GAPDH, since it seems that it increased phospho-prot/total-prot ratio, but lower absolute phospho-prot/loading control. For example, GAPDH level for sample 8 is very low, so one can assume that for example, p-IRF3 and p-TBK1 levels are not changed for this sample. In this regard, the quantifications do not match to the blots. Are there other blots from independent experiments available for this legend? Quantification should be (phospho-TBK1/loading control)/ (total TBK1/loading control).
11. It is not clear from the text how many mice were used for the RNA-seq. How was the GSEA performed from the perspective of bioinformatics analysis? Which gene sets were used for GSEA; from skin? pancreas? or the common 9 genes? The GSEA can be significantly affected by the number of replicates and since the number of differentially expressed genes is very small (9 genes), the significance of the results can be questioned, specifically in Fig. 1b, c. Provide more details and rationale for these analyses and clearly explain the methods used.
12. Fig 1i: To assure that this suggested similar sequence can be bound by p-IRF3, it is important to show a ChIP-qPCR in the presence of IRF3 knock down, noting that the antibody used is not validated for ChIP assays. Also in line 138, how was the sequence achieved?
13. For the GGPP experiment, did the authors monitor HMGCR levels?
14. It is known that 7-dehydrocholesterol reductase (DHCR7) inhibits IRF3 activation in liver tissue. This enzyme catalyzes 7-dehydrocholesterol to cholesterol. Did the authors test if the reduction of IL-33 is also a result of cholesterol reduction?
15. Extended Data Fig. 4 c-e: How many experiments are present in this screening? In panel "d", what are the two "no treat" conditions? And why one is unchanged, but the 2nd is increased? The effect of O16 is compared to the 2nd "no treat", but if compared to 1st there is no change. This needs clarification. In panel "e" the label for poly(I:C) is missing.
16. What are PyMtg breast cells? And why they are used and not a pancreatic cell line for instance?

Minor issues and suggestions:

1. In the introduction, it would be beneficial to mention specific statistics regarding the at-risk populations that require improved cancer prevention strategies. For example, individuals with pancreatitis or other related conditions could be highlighted to underscore the importance of the subject.
2. Line 49: Introduce and define the acronym for TGFbeta
3. Line 63: Clarify the term "damaged tissues" to ensure greater clarity.
4. Line 64: The authors state that IL-33 and its receptor are highly expressed in colitis, pancreatitis, and chronic obstructive pulmonary disease. However, they do not mention the expression of IL-33 in inflammatory bowel disease (IBD) and hepatitis, which are the high-risk populations discussed in the introduction.
5. When discussing the role of IL-33 in cancer development, provide specific examples or studies that support the complex role of IL-33 in promoting or suppressing tumor growth in the high-risk populations under study. This would further underscore the importance of blocking its expression.
6. Line 83: The information about Glutaredoxin-1 is irrelevant as the author does not previously describe it as a trigger for IL-33 activation and they don't explore it further in the paper.
7. What was the positive control(s) for IL-33 decrease that was used in the drug library screen?
8. Line 344, pitavastatin treatment: To ensure accuracy and clarity, I would recommend describing the solubility enhancement method since pitavastatin is not very soluble in water or aqueous solutions like PBS due to its lipophilic nature.
9. Fig. 1e: I suggest changing the GAPDH blot to a clear one in this figure. The bands of mice 6 and 7 are mixed.
10. Extended Data Fig. 3c: How IL-33 levels are changed in WT in this experimental setup? Could be nice to add this on graph side by side other conditions.
11. In line 172, please discriminate cell lines, because the reader can assume both cell lines are

breast.

12. What would be the author's prediction of the signaling they found in an immunodeficient context. Do they think that pitavastatin could further reduce the inflammation in immunocompromised mice/human? This can be added to the discussion.

13. I am curious why induction of Chronic pancreatitis with caerulein is for 6 hours, three days per week for three weeks and for Caerulein-mediated pancreatic cancer is hour for 7 hours, for two consecutive days. They mentioned in the introduction that the upregulation of IL-33 during the transition from acute to chronic inflammation initiates the development of a tumor-promoting immune environment. How did they control the stage of the inflammation in different treatment conditions?

Reviewer #2 (Remarks to the Author):

This is a well written, well executed manuscript that addresses the mechanism of IL33 expression in epithelial cells (pancreas and skin) during chronic inflammation. They authors show that a TLR3-TBK1 axis links chronic inflammation to IL33 expression. They perform a drug library screening and identify pitavastatin as an inhibitor of TBK1 membrane localization and activation. Pitavastatin treatment reduces IL3 expression and protects the pancreas from pancreatitis and cancer formation.

While generally well executed, there are some areas that would benefit from additional attention and can be addressed experimentally or by amending the text.

1) In figure 1, the immunostaining for IL33 appears to show both epithelial and stromal expression (see also point 4). Co-IF with Ecadherin or other epithelial marker would be useful to determine what fraction of IL33 expression is in epithelial cells.

2) The authors refer to Pitavastatin as an IL-33 inhibitor, but it's really a mevalonate inhibitor that has a downstream effect of blocking pTBK1 and thus IL-33 expression. The wording on this oversimplifies the fact that there are undoubtedly many other changes that aren't only IL-33 related. While the language should be clarified, the fact that the inflammation phenotype is blocked in IL-33 knockout tissues (figure S6b&c) partially alleviates this concern.

3) All of the in vitro experiments are only done in a keratinocyte cell line and not in pancreas cells, even as pancreas is the main focus of the in vivo study. Key exp[eriments should be repeated in pancreatic lines.

4) While the authors only refer to "epithelial" IL-33, they are only doing IHC, whole tissue lysate Westerns, and bulk seq, so the evidence that the effects are mediated by epithelial IL-33 is limited. Specifically figure 1A definitely shows stromal cells and not epithelial cells.

5) Literature supporting IL3 expression in the pancreas, both in epithelial and stromal compartments, needs to be cited, such as PMID 32076273, PMID 35245687 and PMID35120601

Reviewer #3 (Remarks to the Author):

In this manuscript, Park et al. investigated the role of IL-33 in chronic inflammation and cancer development and identify an FDA-approved inhibitor of IL-33. They have determined the underlying mechanism. The authors conclude that statin prevents cancer development in chronic inflammation by blocking interleukin 33 expression. Overall, the authors have provided data to support their statements and conclusion. The findings will advance our understanding of chronic inflammation and cancer development. The identified FDA-approved inhibitor has the potential to help patients. I have some comments that may help to improve the manuscript:

1. In the section of results, #1, the author might consider taking "TBK1-IRF3" out from the title, which didn't mention until the next part.
2. In Fig 1d,e, and Fig 3a and their legends, please specify the "NF-kB". Is it the p65? Because there are multiple members of NF-kB.
3. In the results, on page 9, the authors claimed "Pitavastatin suppresses chronic inflammation and its cancer sequela in an IL-33-dependent manner", but Fig 3 did not show "in an IL-33-dependent manner", suggesting moving some of IL33KO data from sub to fig 3.
4. In Fig 3, the authors only showed the tumor weight. Was the survival of the mice checked?
5. What are the DAMPs for chronic inflammation? The authors implied that the S100a8 and S100a9 are potential DAMPs but used PAMPs (LPS and Poly: IC) in the study. Have S100a8 and S100a9 been tested?

RESPONSE TO REVIEWERS' COMMENTS

We thank the reviewers for their insightful comments that have improved our manuscript. Our replies to the comments are provided in blue below. Manuscript text and figures have also been revised (highlighted in blue) according to the reviewers' comments.

Reviewer #1:

Park et. al suggest a novel TLR3-TBK1-IRF3 axis induces IL-33 expression during chronic inflammation in skin and pancreas. The authors perform a FDA-approved drug screen and identify pitavastatin as an inhibitor of IL-33 expression by blocking TBK1 membrane localization and activity. They use in vitro and in vivo assays to show the efficacy of pitavastatin inhibition of skin inflammation and chronic pancreatitis as well as in reducing pancreatic tumor progression. In general, the results are intriguing, however several details are missing, mechanism remains unclear, and references to previous studies requires clarification to address novelty. Major and minor issues are listed below.

We thank the reviewer for their positive remarks and the critical points raised.

Major issues:

1. The mechanism of pitavastatin blocking TBK1 membrane localization remains unclear. As they show this inhibition can be reversed with the addition of exogenous GGPP, the evidence shows GGPP is involved in TBK1 membrane localization and activation, but the mechanism remains unclear. GGPP is important for several processes, including i) protein isoprenylation, ii) production of coenzyme Q (ubiquinone) required for electron transport chain activity, and iii) production of dolichol required for protein N-glycosylation. How does GGPP enable TBK1 membrane localization? GGPP-dependent isoprenylation occurs on CAAX-containing proteins. Is TBK1 such a substrate? If so this would suggest a direct effect, which could be shown by an independent approach using GGPP transferase inhibitors to mimic pitavastatin action. In addition, the inhibitory effect of pitavastatin on TBK1 membrane localization could be functionally uncoupled by targeting TBK1 membrane localization by a mevanlonate pathway- and GGPP-independent method, e.g. myristoylation of TBK1. If not, then what is the mechanism? This is a major gap that needs to be addressed.

We thank the reviewer for raising this critical point. To examine the role of protein isoprenylation in mediating the effect of GGPP on TBK1 signaling, we used 5 μ M GGPP transferase inhibitor (geranylgeranyltransferase I (GGTase-I) inhibitor, GGTI-2147, Millipore Sigma, Catalog no. 345885) to block TBK1 phosphorylation and membrane localization induced by poly(I:C)^{1,2}. We found GGTase-I inhibitor reduced the membrane recruitment and phosphorylation of TBK1 (Fig. R1). This finding indicates that TBK1 signaling is regulated by GGPP through protein isoprenylation. We aim to further examine whether TBK1 is a direct substrate for GGPP-dependent isoprenylation in our future studies. Furthermore, we examined the possibility that TBK1 is a target of myristoylation; however, based on an *in silico* prediction algorithm (<https://web.expasy.org/myristoylator>), TBK1 does not appear to be a target of myristoylation. We have added these data, discussed the relation between isoprenylation with phosphorylation, and stated the limitation of our work in the revised manuscript (Supplementary Fig. 7c and Results section, page 10, first paragraph).

Figure R1. GGTase-I inhibitor blocks TBK1 activation at the membrane.

(Left) Representative immunoblot of p-TBK1, TBK1, Na,K-ATPase, and GAPDH proteins in membrane and cytosol fraction of poly(I:C)-treated Pam212 cells +/- geranylgeranyltransferase I (GGTase-I) inhibitor (GGTI-2147, 5 μ M). Cells were harvested after 6-hour incubation with poly(I:C), pitavastatin and GGTI-2147. (Right) The ratio of membrane-bound p-TBK1 to Na,K-ATPase protein band intensity from the immunoblots ($n=3$ in each group). Graph shows mean + SD, one-way ANOVA.

2. The authors invoke that pitavastatin and zoledronic acid are functioning similarly. Yet the mechanism of these drugs is very different, with pitavastatin targeting HMGCR and zoledronic acid targeting bisphosphonates. This needs to be mentioned in the text.

We have added this information to the revised manuscript (Result section, page 9, first paragraph).

Moreover, to support their claim of similar mechanism of action, they need to show that zoledronic acid also leads to mis-localization of TBK1 from membranes and that exogenous GGPP reverses the activity of zoledronic acid. If zoledronic acid and pitavastatin do not behave similarly in action and response to GGPP, then their distinct mechanisms of action need to be described and put into context for the reader to be able to follow. Are these two different inhibitors of the mevalonate pathway functioning by a similar or different mechanism.

We appreciate this important comment. In our studies, we utilized zoledronic acid as a second class of mevalonate pathway inhibitors that, similar to statins, blocks GGPP production^{3,4}. As suggested, we examined the impact of zoledronic acid on blocking TBK1 phosphorylation and membrane localization induced by poly(I:C) and tested whether exogenous GGPP could reverse this effect. Like pitavastatin, zoledronic acid inhibited the membrane recruitment and phosphorylation of TBK1, which was reversed by exogenous GGPP (Fig. R2). Thus, the two different inhibitors of the mevalonate pathway function in a similar manner to block TBK1 signaling by suppressing GGPP levels in the cells. These findings have been added to the revised manuscript (Supplementary Fig. 6d and Result section, page 9, second paragraph).

Figure R2. Zoledronic acid inhibits TBK1 activation at the membrane.

(Left) Immunoblot of p-TBK1, TBK1, Na,K-ATPase and GAPDH proteins in membrane and cytosol fraction of poly(I:C)-treated Pam212 cells that received zoledronic acid (Zol, 10 μM) alone or in combination with GGPP (3 μM). Cells were harvested after 6-hour incubation with poly(I:C), zoledronic acid and GGPP. (Right) The ratio of membrane-bound p-TBK1 to Na,K-ATPase protein band intensity from the immunoblots (*n*=3 in each group). Graph shows mean + SD, one-way ANOVA.

3. Membrane localization of TBK1 needs to be validated by an independent assay as the high salt concentrations used during membrane extraction often disrupt localization. The usual method of choice is to monitor TBK1 localization using microscopy and image-based analyses, including immunofluorescence and/or by fluorophore tagging of membrane-bound TBK1.

We appreciate this point. For our membrane localization studies, we have utilized a commercially available kit (Pierce Mem-PER plus membrane protein extraction kit, Thermo scientific, catalog no. 89842), which is validated for the detection of membrane-bound proteins and does not rely on high salt concentrations for membrane extraction^{5,6}. Nonetheless, we have examined p-TBK1 localization at the membrane using immunofluorescence cell staining with an endoplasmic reticulum (ER) marker, calreticulin^{7,8}. After the treatment of Pam212 cells with poly(I:C), p-TBK1 was induced and co-localized with calreticulin. We add this data to the revised manuscript (Supplementary Fig. 6c and Result section, page 9, second paragraph and Methods section, page 21, first paragraph).

Figure R3. Poly(I:C)-induced p-TBK1 co-localizes with ER marker, calreticulin.

Representative immunofluorescence images of p-TBK1 and calreticulin in PBS versus poly(I:C)-treated Pam212 cells. Arrows point to co-localization of p-TBK1 and calreticulin in cells' cytoplasm (scale bar: 25 μm).

4. The results are presently impossible to reproduce as many of the key experimental details are missing. Throughout the manuscript the concentration, delivery method and duration of drug treatment need to be provided for each experiment. For example, Figure 2 and Extended data figure 4: Please mention the concentration of pitavastatin, atorvastatin and GGPP used, as well time of treatment. The lack of activity of rosuvastatin may be an uptake issue as it is hydrophobic. This needs to be mentioned in the text as a caveat to data interpretation.

We apologize for the lack of clarity on experimental details. Although the requested information (e.g., concentration of pitavastatin and GGPP) were provided in the Methods section, we have revised the manuscript to provide this information in figure legends as well. Briefly, we used statins and zoledronic acid at 10 μ M concentration for a 6-hour incubation. For exogenous GGPP, we added 3 μ M GGPP together with the inhibitors. We have also added the comment that the lack of rosuvastatin activity in our assays may reflect the hydrophilic nature of this statin (Result section, page 9, first paragraph and Methods section, page 24, third paragraph).

5. The authors have delivered pitavastatin as a topical and intraperitoneal treatment, yet as cholesterol-control agents their FDA-approved delivery is to take them orally. Therefore the authors need to explain why they didn't treat the mice by oral gavage, to mimic the patient experience. They focused on pitavastatin because it is an approved drug, yet they don't use it by the approved route of delivery. This needs to be discussed and justified.

We thank the reviewer for raising this important point. We designed our mouse studies based on the established literature on the intraperitoneal injection of statins in mice⁹⁻¹¹. Although oral gavage is a more accurate representation of route of statin administration in patients, intraperitoneal injection is a safer route compared with oral gavage for repeated administration of drugs in mice. We have clarified this point in the revised manuscript (Methods section, page 17, second paragraph).

6. Line 225: 'demonstrate' needs to be changed to 'support the model that' as the evidence is only correlative in nature. A direct functional relationship has not been supported by the data provided.

As suggested, we have revised this statement in our manuscript (Results section, page 11, second paragraph).

7. Lines 282 and 283: "Blocking the mevalonate pathway product, GGPP, inhibits cancer cells' amino acid uptake by regulating micropinocytosis (ref 66)." This is incomplete as GGPP is important for many downstream processes (see #1 above), which need to be described in the text and addressed mechanistically.

We have revised this statement in our manuscript according to the reviewer's comments and the new data described in Figure R1 above (Results section, page 10, first paragraph and Discussion section, page 14, third paragraph).

8. Lines 289-291: Overstated. The data provided do not explain mechanism of action.

We have revised this statement to avoid overstatement in our manuscript accordingly (Discussion section, page 15, first paragraph).

9. This axis of IL-33 regulation and the inhibitory role of statins has been previously reported, yet references were not included, e.g. PMID: 36711701; PMID: 31810599. Please include these and similar previous references and clearly delineate to the reader what was previously described and what's new in this report.

We thank the reviewer for pointing this out. Reference 36711701 is actually a pre-print of our manuscript posted on Research Square by Nature Journal during our initial submission. We have added previous publication on IL-33 regulation^{12,13} and clarified the contribution of our work to the field in the revised manuscript (Discussion section, page 14, second paragraph).

10. Figure 1D: Please provide the full membrane for the western blot. The image quality looks dissimilar in different parts of the same membrane. Furthermore, the authors show the ratio of p-TBK1/TBK1 and p-IRF3/IRF3 quantified from immunoblot. However, they should quantify also related to GAPDH, since it seems that it increased phospho-prot/total-prot ratio, but lower absolute phospho-prot/loading control. For example, GAPDH level for sample 8 is very low, so one can assume that for example, p-IRF3 and p-TBK1 levels are not changed for this sample. In this regard, the quantifications do not match to the blots. Are there other blots from independent experiments available for this legend? Quantification should be (phospho-TBK1/loading control) / (total TBK1/loading control).

We appreciate this point. As requested, we provide raw data with a full-blot of each band included in the revised manuscript. With regard to the protein ratios shown in the study, we certainly understand reviewer's point of view. We did not observe any significant change in endogenous protein/loading control ratios (e.g., TBK1/GAPDH and IRF3/GAPDH) between test and control samples (Fig. R4). Hence, we chose to present phospho-protein/endogenous protein ratios to effectively compare the degree of protein phosphorylation in each group. We have added this information to our revised manuscript (Supplementary Fig.3a, b and Methods section, page 20, second paragraph). Please also see Figure R8 below.

Figure R4. Western blot band intensity ratio of endogenous protein to GAPDH loading control.

(Left) Immunoblot of p-TBK1, p-IRF3, p-NF-kB, TBK1, IRF3, NF-kB and GAPDH proteins in DNFB- versus acetone-treated WT skin (n=4 in each group). (Right) The ratio of TBK1/GAPDH and IRF3/GAPDH protein band intensity quantified from the immunoblot shown on the left.

11. It is not clear from the text how many mice were used for the RNA-seq. How was the GSEA performed from the perspective of bioinformatics analysis? Which gene sets were used for GSEA; from skin? pancreas? or the common 9 genes? The GSEA can be significantly affected by the number of replicates and since the number of differentially expressed genes is very small (9 genes), the significance of the results can be questioned, specifically in Fig. 1b, c. Provide more details and rationale for these analyses and clearly explain the methods used.

For RNA-Seq analysis, we used 5 mice in each group (acetone-treated skin, DNFB-treated skin, PBS-treated pancreas, and caerulein-treated pancreas). We performed GSEA on differentially expressed genes between caerulein- and PBS-treated pancreas samples. We have revised the manuscript to clearly state the RNA-Seq parameters in the Methods and Figure legends (Figure 1 legend and Methods section, page 22, second paragraph and page 23, first paragraph).

12. Fig 1i: To assure that this suggested similar sequence can be bound by p-IRF3, it is important to show a ChIP-qPCR in the presence of IRF3 knock down, noting that the antibody used is not validated for ChIP assays. Also in line 138, how was the sequence achieved?

This is an important point. As suggested, we performed ChIP-qPCR on Pam212 cells treated with poly(I:C) after knocking down IRF3, which resulted in the loss of //33 promoter pull-down by anti-pIRF3 antibody (Fig. R5). //33 promoter sequence was obtained from GeneCopoeia, Inc. that provides //33 promoter region clones (<https://www.genecopoeia.com/product/search/detail.php?prt=22&cid=&key=MPRM34949&type=promoter&choose=il33>). These data and information have been added to the revised manuscript (Supplementary Fig. 3m and Results section, page 7, second paragraph and Methods section, page 24, first paragraph).

Figure R5. IRF3 knockdown blocks //33 promoter pull-down by anti-pIRF3 antibody in poly(I:C)-treated cells.

ChIP-qPCR assay for p-IRF3 binding to //33 promoter region after silrf3-treated Pam212 cells exposed to poly(I:C) versus PBS ($n=4$ in each group).

13. For the GGPP experiment, did the authors monitor HMGCR levels?

HMGCR expression was not changed in the GGPP experiment, which means GGPP findings were not related to HMGCR expression level (Fig. R6). This data has been added to the revised manuscript (Supplementary Fig. 7a and Results section, page 10, first paragraph).

Figure R6. HMG-CoA reductase (HMGCR) expression is not altered by treatments that modulate IL-33 expression in keratinocytes.

Hmgcr expression in PBS- and poly(I:C)-treated Pam212 cells that received pitavastatin (10 μM) alone or in combination with GGPP (3 μM) (*n*=4 in each group). Cells were harvested after 6-hour incubation with poly(I:C), pitavastatin and GGPP.

14. It is known that 7-dehydrocholesterol reductase (DHCR7) Inhibits IRF3 activation in liver tissue. This enzyme catalyzes 7-dehydrocholesterol to cholesterol. Did the authors test if the reduction of IL-33 is also a result of cholesterol reduction?

We thank the reviewer for this critical comment. To directly investigate the impact of cholesterol on IL-33 suppression by statins, we treated Pam212 cells with poly(I:C) and pitavastatin in the presence exogenous cholesterol. The addition of exogenous cholesterol did not reverse pitavastatin ability to suppress poly(I:C)-induced IL-33 expression in Pam212 cells (Fig. R7). These findings have been added to the revised manuscript (Supplementary Fig. 7b and Results section, page 10, first paragraph).

Figure R7. //33 suppression by pitavastatin is not affected by exogenous cholesterol.

//33 expression in poly(I:C)-treated Pam212 cells that received pitavastatin (10μM) alone or in combination with cholesterol (5μg/mL) (*n*=4 in each group). Cells were harvested after 6-hour incubation with poly(I:C), pitavastatin and cholesterol.

15. Extended Data Fig. 4 c-e: How many experiments are present in this screening? In panel “d”, what are the two “no treat” conditions? And why one is unchanged, but the 2nd is increased? The effect of O16 is compared to the 2nd “no treat”, but if compared to 1st there is no change. This needs clarification. In panel “e” the label for poly(I:C) is missing.

We performed the screening assay in duplicate (Supplementary Fig. 5c). To confirm hits from screening, we first validated the impact of select candidate on the inhibition of //33 expression induced by poly(I:C) in Pam212 cells (Supplementary Fig. 5d). The two “no treat” controls in the graph denote no inhibitor treatment conditions. However, first “no treat” condition was treated with PBS versus second “no treat” was treated with poly(I:C). Next, we examined the impact of the

select candidates on *IL33* expression in *PyMt^{tg}* breast cancer cell line because it has high endogenous expression of IL-33 and does not require poly(I:C) treatment (Supplementary Fig. 5e). We have clarified these points by adding labels to the figure panels and revising Supplementary Fig. 5 legend.

16. What are *PyMt^{tg}* breast cells? And why they are used and not a pancreatic cell line for instance?

PyMt^{tg} breast cancer cell line was used for testing candidate IL-33 inhibitors because it has high endogenous expression of IL-33 and does not require poly(I:C) treatment. We used this cell line as an independent system to validate the efficacy of candidate IL-33 inhibitors. Nonetheless, we appreciate the point regarding the inclusion of a pancreatic cell line in our studies. To address this point, we performed the GGPP experiment in a pancreas cell line, 839WT, which our collaborators have generated from an organoid culture of pancreas from a wild-type mouse on the C57BL/6 background. Similar to Pam212 cells, poly(I:C) treatment induced TBK1 membrane localization and phosphorylation in 839WT pancreatic cells, which was suppressed by pitavastatin and reversed by the addition of exogenous GGPP (Fig. R8). These data have been added to the revised manuscript (Supplementary Fig. 6a, b, and Results section, page 9, second paragraph).

Figure R8. Pitavastatin inhibits TBK1 activation by regulating GGPP in a pancreatic cell line.

a (Left) Immunoblot of p-TBK1, p-IRF3, TBK1, IRF3 and GAPDH proteins in whole cell lysates of poly(I:C)-treated 839WT cells that received pitavastatin (10 μ M) alone or in combination with GGPP (3 μ M) ($n=3$ in each group). Cells were harvested after 6-hour incubation with poly(I:C), pitavastatin and GGPP. (Right) The ratio of p-TBK1/TBK1, p-IRF3/IRF3, TBK1/GAPDH, and IRF3/GAPDH protein band intensity from immunoblots ($n=3$ in each group). **b** (Left) Immunoblot of p-TBK1, TBK1, Na,K-ATPase, and GAPDH proteins in membrane and cytosol fraction of poly(I:C)-treated 839WT cells that received pitavastatin (10 μ M) alone or in combination with GGPP (3 μ M). Cells were harvested after 6-hour incubation with poly(I:C), pitavastatin and GGPP. (Right) The ratio of membrane-bound p-TBK1/Na,K-ATPase protein band intensity from the immunoblots ($n=3$ in each group).

Minor issues and suggestions:

1. In the introduction, it would be beneficial to mention specific statistics regarding the at-risk populations that require improved cancer prevention strategies. For example, individuals with pancreatitis or other related conditions could be highlighted to underscore the importance of the subject.

We have revised our manuscript to address this point (Introduction section, page 3, first paragraph).

2. Line 49: Introduce and define the acronym for TGFbeta

We have revised the manuscript as recommended (Introduction section, page 3, first paragraph).

3. Line 63: Clarify the term "damaged tissues" to ensure greater clarity.

We have revised the manuscript as recommended (Introduction section, page 3, second paragraph).

4. Line 64: The authors state that IL-33 and its receptor are highly expressed in colitis, pancreatitis, and chronic obstructive pulmonary disease. However, they do not mention the expression of IL-33 in inflammatory bowel disease (IBD) and hepatitis, which are the high-risk populations discussed in the introduction.

We have revised the manuscript to include IBD and hepatitis references as recommended (Introduction section, page 3, second paragraph).

5. When discussing the role of IL-33 in cancer development, provide specific examples or studies that support the complex role of IL-33 in promoting or suppressing tumor growth in the high-risk populations under study. This would further underscore the importance of blocking its expression.

We have revised the manuscript as recommended (Discussion section, page 13, second paragraph).

6. Line 83: The information about Glutaredoxin-1 is irrelevant as the author does not previously describe it as a trigger for IL-33 activation and they don't explore it further in the paper.

We thank the reviewer for pointing this out. We have revised the manuscript accordingly.

7. What was the positive control(s) for IL-33 decrease that was used in the drug library screen?

We have used a TBK1 inhibitor, BX795, as positive control in IL-33 suppression assays. We have added this information to the revised manuscript (Methods section, page 19, first paragraph).

8. Line 344, pitavastatin treatment: To ensure accuracy and clarity, I would recommend describing the solubility enhancement method since pitavastatin is not very soluble in water or aqueous solutions like PBS due to its lipophilic nature.

As recommended in pitavastatin datasheet, we used DMSO to dissolve pitavastatin and generate a high concentration stock solution. Pitavastatin stock solution was diluted in PBS for use in mouse experiments. DMSO with PBS solution was used for the control group. This information has been added to the revised manuscript (Methods section, page 17, second paragraph).

9. Fig. 1e: I suggest changing the GAPDH blot to a clear one in this figure. The bands of mice 6 and 7 are mixed.

We have revised Fig. 1e accordingly.

10. Extended Data Fig. 3c: How IL-33 levels are changed in WT in this experimental setup? Could be nice to add this on graph side by side other conditions.

IL-33 level in DNFB-treated WT skin is shown in Supplementary Fig. 2b. As suggested, we have added this group to Supplementary Fig. 4c to help with side-by-side comparison.

11. In line 172, please discriminate cell lines, because the reader can assume both cell lines are breast.

We have revised the manuscript as recommended (Results section, page 8, second paragraph).

12. What would be the author's prediction of the signaling they found in an immunodeficient context. Do they think that pitavastatin could further reduce the inflammation in immunocompromised mice/human? This can be added to the discussion.

We thank the reviewer for raising this interesting point. Considering the tumor promoting function of nuclear IL-33, we anticipate statins will exert a cancer-suppressive effect in chronic inflammation in the immunocompromised settings. We have discussed this point in the revised manuscript (Discussion section, page 13-14, second paragraph).

13. I am curious why induction of Chronic pancreatitis with caerulein is for 6 hours, three days per week for three weeks and for Caerulein-mediated pancreatic cancer is hour for 7 hours, for two consecutive days. They mentioned in the introduction that the upregulation of IL-33 during the transition from acute to chronic inflammation initiates the development of a tumor-promoting immune environment. How did they control the stage of the inflammation in different treatment conditions?

In our studies, we followed the established methods to induce chronic pancreatitis in WT mice versus initiate tumor development in the context of chronic inflammation in KPC animals¹⁴⁻¹⁸. We confirmed that in both experimental settings, IL-33 was highly upregulated, which was inhibited by pitavastatin in our studies. We have clarified this point in our revised manuscript by providing additional references (Results section, page 6, first paragraph).

Reviewer #2:

This is a well written, well executed manuscript that addresses the mechanism of IL33 expression in epithelial cells (pancreas and skin) during chronic inflammation. They authors show that a TLR3-TBK1 axis links chronic inflammation to IL33 expression. They perform a drug library screening and identify pitavastatin as an inhibitor of TBK1 membrane localization and activation. Pitavastatin treatment reduces IL3 expression and protects the pancreas from pancreatitis and cancer formation. While generally well executed, there are some areas that would benefit from additional attention and can be addressed experimentally or by amending the text.

We thank the reviewer for their positive remarks and the important points raised.

1) In figure 1, the immunostaining for IL33 appears to show both epithelial and stromal expression (see also point 4). Co-IF with Ecadherin or other epithelial marker would be useful to determine what fraction of IL33 expression is in epithelial cells.

We thank the reviewer for raising this point. As suggested, we performed Co-IF with IL-33 and epithelial marker 'cytokeratin' in chronic skin and pancreas inflammation. IL-33 was expressed in cytokeratin-positive epithelial cells in the skin and pancreas (Fig. R9). We have added this data to our revised manuscript (Supplementary Fig. 2e, Supplementary Fig. 9f, Results section, page 6, first paragraph, and Results section, page 11, second paragraph).

Figure R9. IL-33 is overexpressed in epithelial cells in chronic skin and pancreas inflammation. Representative immunofluorescence images of IL-33 and cytokeratin-stained inflamed skin and pancreas (scale bar: 100 μ m).

2) The authors refer to Pitavastatin as an IL-33 inhibitor, but it's really a mevalonate inhibitor that has a downstream effect of blocking pTBK1 and thus IL-33 expression. The wording on this oversimplifies the fact that there are undoubtedly many other changes that aren't only IL-33 related. While the language should be clarified, the fact that the inflammation phenotype is blocked in IL-33 knockout tissues (figure S6b&c) partially alleviates this concern.

As suggested, we have revised our manuscript to avoid oversimplifying the effect of pitavastatin. (Abstract, page 2 and Introduction section, page 4, third paragraph).

3) All of the in vitro experiments are only done in a keratinocyte cell line and not in pancreas cells, even as pancreas is the main focus of the in vivo study. Key experiments should be repeated in pancreatic lines.

We appreciate this point. As suggested, we have performed pitavastatin/GGPP study in a pancreas cell line, 839WT (please refer to response to Reviewer #1 above). Similar to Pam212 cells, poly(I:C) treatment induced TBK1 membrane localization and phosphorylation in 839WT pancreatic cells, which was suppressed by pitavastatin and reversed by the addition of exogenous GGPP (Fig. R8 above). These data have been added to the revised manuscript (Supplementary Fig. 6a, b, and Results section, page 9, second paragraph).

4) While the authors only refer to "epithelial" IL-33, they are only doing IHC, whole tissue lysate Westerns, and bulk seq, so the evidence that the effects are mediated by epithelial IL-33 is limited. Specifically figure 1A definitely shows stromal cells and not epithelial cells.

Please refer to response to comment #1 above. We certainly agree that IL-33 can be expressed by stromal cells in chronic inflammation; however, we find that IL-33 is also overexpressed in epithelial cells in chronic inflammation. Thus, blocking IL-33 expression, instead of blocking its cytokine function alone, is key to the optimal blockade of IL-33 signaling for cancer prevention in chronic inflammation. We have added this information to our revised manuscript (Supplementary Fig. 2e and Results section, page 6, first paragraph).

5) Literature supporting IL-33 expression in the pancreas, both in epithelial and stromal compartments, needs to be cited, such as PMID 32076273, PMID 35245687 and PMID35120601

We have cited the recommended references in our revised manuscript (Discussion section, page 13, second paragraph).

Reviewer #3:

In this manuscript, Park et al. investigated the role of IL-33 in chronic inflammation and cancer development and identify an FDA-approved inhibitor of IL-33. They have determined the underlying mechanism. The authors conclude that statin prevents cancer development in chronic inflammation by blocking interleukin 33 expression. Overall, the authors have provided data to support their statements and conclusion. The findings will advance our understanding of chronic inflammation and cancer development. The identified FDA-approved inhibitor has the potential to help patients. I have some comments that may help to improve the manuscript:

We thank the reviewer for their constructive comments.

1. In the section of results, #1, the author might consider taking “TBK1-IRF3” out from the title, which didn't mention until the next part.

We thank the reviewer for pointing out this mistake, which we have been corrected in the revised manuscript.

2. In Fig 1d,e, and Fig 3a and their legends, please specify the “NF-kB”. Is it the p65? Because there are multiple members of NF-kB.

Yes, the NF-κB referenced in the manuscript is p65. We have clarified this in our revised manuscript (Fig. 1 and Result section, page 7, first paragraph)

3. In the results, on page 9, the authors claimed “Pitavastatin suppresses chronic inflammation and its cancer sequela in an IL-33-dependent manner”, but Fig 3 did not show “in an IL-33-dependent manner”, suggesting moving some of IL33KO data from sub to fig 3.

As suggested, we have moved IL-33^{KO} data from Supplementary Fig. 6 to Fig. 3 in the revised manuscript.

4. In Fig 3, the authors only showed the tumor weight. Was the survival of the mice checked?

We thank the reviewer for raising this critical point. In our studies, we have focused on identifying mechanism and therapies that **prevent** cancer development in chronic inflammation. As such, the pancreatitis-associated pancreatic cancer model we used (i.e., KPC plus caerulein) is a suitable model to interrogate statin's impact on early pancreatic cancer development. Nonetheless, as suggested, we have studied survival rate of caerulein-exposed KPC mice treated with pitavastatin versus vehicle control (Fig. R10a). Pitavastatin treatment increased the survival rate of KPC mice compared with control-treated animals (Fig. R10b). Moreover, pitavastatin treatment retained the pancreatic tumors in a pre-cancerous stage (Fig. R10c). We have added this data to our revised manuscript (Fig 3h and Supplementary Fig. 9g, h, and Results section, page 11, second paragraph).

Fig. R10. Pitavastatin increases the survival of KPC mice affected by caerulein-induced chronic pancreatitis and pancreatic cancer.

a Schematic diagram of the experimental design for the induction of chronic inflammation-associated pancreas cancer in mice for survival analysis. KPC mice received seven hourly intraperitoneal caerulein injections over two days. Next, mice received intraperitoneal pitavastatin versus PBS injections once every three days over four weeks followed by weekly treatments until the animal reached a moribund stage. **b** Survival of caerulein-exposed KPC mice treated with pitavastatin ($n=6$) versus PBS ($n=5$, log-rank test). **c** Representative images of H&E and Alcian blue-stained terminal pancreatic tumors from pitavastatin- versus PBS-treated KPC mice that underwent caerulein-induced pancreatic cancer protocol.

5. What are the DAMPs for chronic inflammation? The authors implied that the S100a8 and S100a9 are potential DAMPs but used PAMPs (LPS and Poly: IC) in the study. Have S100a8 and S100a9 been tested?

We appreciate this comment. We identified S100a8 and S100a9 as potential DAMPs that activate IL-33 *in vivo*. In the same analysis, we confirmed TLR3/4 pathways to be activated in chronic inflammation, which are the known receptors for S100a8 and S100a9^{19,20}. Therefore, in our *in vitro* studies, we utilized poly(I:C) and LPS as established agents used in the field to activate TLR3/4 pathways so to discover downstream factors linking TLR activation to IL-33 expression^{12,21}. Hence, we propose that poly(I:C) and LPS are suitable agents as specific TLR3/4

activators for *in vitro* studies. We have clarified this point in the revised manuscript (Result section, page 7, first paragraph).

References

- 1 Subasinghe, W., Syed, I. & Kowluru, A. Phagocyte-like NADPH oxidase promotes cytokine-induced mitochondrial dysfunction in pancreatic beta-cells: evidence for regulation by Rac1. *Am J Physiol Regul Integr Comp Physiol* **300**, R12-20, doi:10.1152/ajpregu.00421.2010 (2011).
- 2 Chang, S. Y. *et al.* Inhibitors of protein geranylgeranyltransferase-I lead to prelamin A accumulation in cells by inhibiting ZMPSTE24. *J Lipid Res* **53**, 1176-1182, doi:10.1194/jlr.M026161 (2012).
- 3 Goffinet, M. *et al.* Zoledronic acid treatment impairs protein geranyl-geranylation for biological effects in prostatic cells. *BMC Cancer* **6**, 60, doi:10.1186/1471-2407-6-60 (2006).
- 4 Sethunath, V. *et al.* Targeting the Mevalonate Pathway to Overcome Acquired Anti-HER2 Treatment Resistance in Breast Cancer. *Mol Cancer Res* **17**, 2318-2330, doi:10.1158/1541-7786.MCR-19-0756 (2019).
- 5 Zhao, Y. *et al.* Ubl4A is required for insulin-induced Akt plasma membrane translocation through promotion of Arp2/3-dependent actin branching. *Proc Natl Acad Sci U S A* **112**, 9644-9649, doi:10.1073/pnas.1508647112 (2015).
- 6 Phuchareon, J., McCormick, F., Eisele, D. W. & Tetsu, O. EGFR inhibition evokes innate drug resistance in lung cancer cells by preventing Akt activity and thus inactivating Ets-1 function. *Proc Natl Acad Sci U S A* **112**, E3855-3863, doi:10.1073/pnas.1510733112 (2015).
- 7 Muller-Taubenberger, A. *et al.* Calreticulin and calnexin in the endoplasmic reticulum are important for phagocytosis. *EMBO J* **20**, 6772-6782, doi:10.1093/emboj/20.23.6772 (2001).
- 8 Ni Fhlathartaigh, M. *et al.* Calreticulin and other components of endoplasmic reticulum stress in rat and human inflammatory demyelination. *Acta Neuropathol Commun* **1**, 37, doi:10.1186/2051-5960-1-37 (2013).
- 9 Merx, M. W. *et al.* Statin treatment after onset of sepsis in a murine model improves survival. *Circulation* **112**, 117-124, doi:10.1161/CIRCULATIONAHA.104.502195 (2005).
- 10 Beffa, D. C. *et al.* Simvastatin treatment improves survival in a murine model of burn sepsis: Role of interleukin 6. *Burns* **37**, 222-226, doi:10.1016/j.burns.2010.10.010 (2011).
- 11 Altintas, N. D., Atilla, P., Iskit, A. B. & Topeli, A. Long-term simvastatin attenuates lung injury and oxidative stress in murine acute lung injury models induced by oleic Acid and endotoxin. *Respir Care* **56**, 1156-1163, doi:10.4187/respcare.00770 (2011).
- 12 Natarajan, C., Yao, S. Y. & Sriram, S. TLR3 Agonist Poly-IC Induces IL-33 and Promotes Myelin Repair. *PLoS One* **11**, e0152163, doi:10.1371/journal.pone.0152163 (2016).
- 13 Jin, M., Komine, M., Tsuda, H., Oshio, T. & Ohtsuki, M. dsRNA induces IL-33 promoter activity through TLR3-EGFR-IRF3 pathway in normal human epidermal keratinocytes. *J Dermatol Sci* **96**, 178-180, doi:10.1016/j.jdermsci.2019.11.002 (2019).
- 14 Lin, W. R. *et al.* Granulocyte colony-stimulating factor reduces fibrosis in a mouse model of chronic pancreatitis. *PLoS One* **9**, e116229, doi:10.1371/journal.pone.0116229 (2014).
- 15 Murakami, S. *et al.* Yes-associated protein mediates immune reprogramming in pancreatic ductal adenocarcinoma. *Oncogene* **36**, 1232-1244, doi:10.1038/nc.2016.288 (2017).
- 16 Kong, B. *et al.* Dynamic landscape of pancreatic carcinogenesis reveals early molecular networks of malignancy. *Gut* **67**, 146-156, doi:10.1136/gutjnl-2015-310913 (2018).
- 17 Komar, H. M. *et al.* Inhibition of Jak/STAT signaling reduces the activation of pancreatic stellate cells in vitro and limits caerulein-induced chronic pancreatitis in vivo. *Sci Rep* **7**, 1787, doi:10.1038/s41598-017-01973-0 (2017).
- 18 Chen, S. M., Chieng, W. W., Huang, S. W., Hsu, L. J. & Jan, M. S. The synergistic tumor growth-inhibitory effect of probiotic *Lactobacillus* on transgenic mouse model of pancreatic

- cancer treated with gemcitabine. *Sci Rep* **10**, 20319, doi:10.1038/s41598-020-77322-5 (2020).
- 19 Tsai, S. Y. *et al.* Regulation of TLR3 Activation by S100A9. *J Immunol* **195**, 4426-4437, doi:10.4049/jimmunol.1500378 (2015).
- 20 Guo, Q. *et al.* Induction of alarmin S100A8/A9 mediates activation of aberrant neutrophils in the pathogenesis of COVID-19. *Cell Host Microbe* **29**, 222-235 e224, doi:10.1016/j.chom.2020.12.016 (2021).
- 21 Weinberg, E. O. *et al.* IL-33 induction and signaling are controlled by glutaredoxin-1 in mouse macrophages. *PLoS One* **14**, e0210827, doi:10.1371/journal.pone.0210827 (2019).

REVIEWERS' COMMENTS

Reviewer #1 (Remarks to the Author):

The authors have been highly responsive to reviewers comments, which has added clarity and balance to the manuscript.

Reviewer #3 (Remarks to the Author):

Thank you for doing a great job of addressing the reviewers' comments.

RESPONSE TO REVIEWERS' COMMENTS

Reviewer #1:

The authors have been highly responsive to reviewers' comments, which has added clarity and balance to the manuscript.

We thank the reviewer for their positive remarks.

Reviewer #3:

Thank you for doing a great job of addressing the reviewers' comments.

We thank the reviewer for their positive remarks.